# Dietary Polyphenols in Metabolic and Neurodegenerative Diseases: Molecular Targets in Autophagy and Biological Effects

**DOI:** 10.3390/antiox10020142

**Published:** 2021-01-20

**Authors:** Ana García-Aguilar, Olga Palomino, Manuel Benito, Carlos Guillén

**Affiliations:** 1Department of Pharmacology, Pharmacognosy and Botany, Faculty of Pharmacy, University Complutense of Madrid, Ciudad Universitaria s/n, 28040 Madrid, Spain; ana.garcia.aguilar@ucm.es (A.G.-A.); olgapalomino@farm.ucm.es (O.P.); 2Department of Biochemistry and Molecular Biology, Faculty of Pharmacy, Complutense University of Madrid, Ciudad Universitaria s/n, 28040 Madrid, Spain; mbenito@ucm.es; 3Spanish Biomedical Research Centre in Diabetes and Associated Metabolic Disorders (CIBERDEM), Instituto de Salud Carlos III, 28040 Madrid, Spain

**Keywords:** polyphenols, autophagy, mechanisms, oxidative stress, inflammation, disease models

## Abstract

Polyphenols represent a group of secondary metabolites of plants which have been analyzed as potent regulators of multiple biological processes, including cell proliferation, apoptosis, and autophagy, among others. These natural compounds exhibit beneficial effects and protection against inflammation, oxidative stress, and related injuries including metabolic diseases, such as cardiovascular damage, obesity and diabetes, and neurodegeneration. This review aims to summarize the mechanisms of action of polyphenols in relation to the activation of autophagy, stimulation of mitochondrial function and antioxidant defenses, attenuation of oxidative stress, and reduction in cell apoptosis, which may be responsible of the health promoting properties of these compounds.

## 1. Introduction

One of the consequences of the normal function in living organisms is the intracellular production of reactive oxygen (ROS) and nitrogen (RNS) species in significant amounts, mainly found in the cytosol, mitochondria, lysosomes, peroxisomes, and epithelial membranes. The deleterious effects of free radicals are counteracted by the antioxidant defenses present at cellular level through both enzymatic and non-enzymatic systems, leading to the concept of redox homeostasis [1]. However, when the antioxidant defenses are not able to counteract an extensive free radicals production, oxidative stress occurs by reacting with lipids, proteins, and nucleotides to produce oxidized compounds, which are the final responsible of the cell damage [2,3]. These processes are mediated by several pathways implying signal transduction such as mitochondrial kinases, protein kinase A, protein kinase B/Akt, protein kinase C (PKC), extracellular signal-regulated protein kinase, c-Jun N-terminal kinase, or p38 nitrogen activated kinase [4,5,6,7,8,9].

The main response to tissue damage caused by intense oxidative stress is cell death. The most studied type of cell death is apoptosis. This process runs through several steps, starting with the nucleus condensation and fragmentation; mitochondrial swelling and destruction of the endoplasmic reticulum (ER) structure are then observed, together with cytoplasmic vacuolization and the disappearance of the microvilli. All these changes finally cause cell contraction and death, without any immune reaction.

Moreover, in response to cellular stress, eukaryotic cells can activate several physiological degradative pathways, such as the ubiquitin (Ub)-proteasome system (UPS) and autophagy.

Autophagy is a lysosomal degradation pathway for clearing out cytoplasmic long-lived proteins and damaged organelles [10], to mitigate cellular stress, and to increase the chances of cell survival [11]. Although oxidative stress stimulates autophagy in different cellular systems, it has been shown that a prolonged oxidative stress status inhibits autophagy [12,13]. Autophagy dysfunction leads to the accumulation of protein aggregates, oxidative and ER stress, which in turn contribute to cell apoptosis and the progression of several diseases [14,15]. Although autophagy can promote cell death under certain circumstances, it is considered as a protective mechanism. Furthermore, both positive and negative regulatory loops have been described for different proteins involved in either apoptosis or autophagy processes [16,17].

As a result of an increased oxidative stress and depletion of intracellular antioxidants, additional exogenous sources of counteracting molecules such as fat-soluble vitamin E, beta-carotene, coenzyme Q, or water-soluble vitamin C, are needed. Increasing evidence demonstrates that dietary polyphenols act as antioxidants and autophagy activators, thus reducing oxidative stress [12]. Polyphenols constitute one of the main groups of secondary metabolites from plants. Their role in the vegetal cell metabolism involves the cellular defense against ultraviolet radiation or aggression by pathogens, whereas they contribute to the bitterness, astringency, color, flavor, odor, and oxidative stability of the plant. 

More than 8000 polyphenolic structures have been identified in plants with a wide range of molecular size depending on the number of phenolic rings and the nature and the number of the bound chemical groups, leading to simple compounds such as phenolic acids or highly polymerized molecules as tannins. The first classification system grouped natural compounds according to their chemical structure, so compounds containing at least one phenolic group could be mainly found among heterosides and tannins [18]. The biogenetic approach adopted more than 30 years ago allowed a new classification of metabolites from plants according to their main biogenetic route, leading to a division of phenolic compounds derived from shikimate (shikimic acid pathway) or acetate, with rather different biochemical and biological characteristics [18]. Thus, polyphenols derived from the shikimic acid pathway (mainly phenolic acids, coumarins, lignans, flavonoids, and anthocyanins) show prominent pharmacological activities such as antioxidant, antiinflammatory, and antiproliferative effects which led to an increased interest in the study of these compounds [19,20,21,22,23] (Table 1). Among them, flavonoids and the seven described subclasses (flavonols, flavones, flavanones, flavanonols, flavanols, anthocyanidins, and isoflavones) are the most abundant and studied group of polyphenols, although the exact mechanisms involved in the health-promoting effects of these compounds are still far from clear. Figure 1 shows the main types of polyphenols with biological interest. 

The direct antioxidant activity of polyphenols is mainly due to their redox properties which allow them to act as chelating agents of metal ions, hydrogen donators, or reducing agents. 

Nevertheless, caution is needed because its total antioxidant capacity measured by in vitro tests does not reflect the real potential of substances and its kinetics does not always allow the proximity of a reductant or an antioxidant to the oxidative injury so that the redox reaction could take place [24,25].

At molecular level, polyphenols, rather than acting as merely chemical antioxidants, may induce enzymatic systems which in turn alter the steady-state levels of protective elements, such as catalase (CAT), superoxide dismutase (SOD), glutathione peroxidase (GPx), and heme oxygenase 1 (OH1), and notably protect cells against the oxidative insult [26,27].

Other polyphenols such as hydroxycinnamic acids (i.e., caffeic and ferulic acids) prevent peroxidation mediated by UVR by inhibiting the chain reaction of lipid peroxidation and scavenging nitrogen oxides; both compounds strongly absorb UV-photons, leading to a protective effect, i.e., on human skin from UVB-induced erythema which justifies their inclusion in the formulation of several topical solutions and sunscreens [28].

More recent studies are focused in the indirect mechanisms of the antioxidant activity which are more difficult to measure but include hormesis, defined as the ability to upregulate those enzymes needed in innate detox pathways and/or the transcription of the so-called vitagenes, which in turn improve the innate mechanisms of detoxification [25]; moreover, their activity in the adaptive cellular response, genes transcription [26,29], and even in autophagic process as an important cellular mechanism underlying the beneficial effects of some polyphenols have been described [30,31].

In humans, most polyphenols are ingested from the diet (fruits—especially red berries—, vegetables, cereals, and beverages) in the form of glycosides, which are bound to one or more sugar molecules [32]. After ingestion, glycosides are hydrolyzed by intestinal hydrolases of gut microflora and the released aglycones are then absorbed by the intestinal cells and converted into their respective metabolites in both intestinal and hepatic cells. Metabolites are then transported by the blood to various cells and/or excreted by urine.

The oral bioavailability of dietary polyphenols is low due to their poor absorption, rapid excretion, and extensive biotransformation and conjugation that take place during their absorption from the intestine and at hepatic level [25,33].

Epidemiological data suggested that long-term consumption of a polyphenol-rich diet is associated with lower risk of developing certain cancers and coronary heart diseases, a decrease in arterial pressure and plasma concentration of lipids, as well as a protection against development of diabetes, osteoporosis, and even neurodegenerative diseases. Nonetheless, overconsumption of some types of food has been associated with the induction of oxidative stress and the consequent development of pathological transformations [27,34].

The aim of this review is to contribute to the knowledge of the main mechanisms of action of dietary polyphenols in relation to autophagy regulation and their impact on several metabolic diseases in order to allow the development of new effective drugs for the treatment of oxidative stress-related diseases.

## 2. Molecular Mechanisms of Autophagic Process and Its Regulation by Dietary Polyphenols

Autophagy is a physiological recycling process highly conserved in eukaryotic cells, playing an important role in maintaining cytoplasmic quality control through the elimination of cytosolic protein aggregates, damaged organelles, and invasive microbes and reducing oxidative and ER stress. This activity aims to maintain cellular homeostasis and cell survival, which contributes to enhance health and longevity [35,36,37,38]. Moreover, this catabolic process is upregulated by a wide range of extracellular and intracellular stressors such as nutrient starvation including growth factor and insulin deprivation, hypoxia, infections, ER, and oxidative stress; the resulting macromolecular constituents are then recycled. In this context, activation of autophagy allows metabolic adaptation during starvation by providing an alternative source of energy and nutrients [39] and participating in the cellular defense against infections [40,41]. The autophagic process involves the activities of proteins encoded by autophagy-related (*ATG*) genes and it could be considered as a nonselective degradative pathway, removing non-specific cargoes and ubiquitinated proteins (also referred as “bulk autophagy”), or highly specific, eliminating distinct cargoes including lipids (termed lipophagy), peroxisomes (pexophagy), mitochondria (mitophagy), nucleus (nucleophagy), ribosomes (ribophagy), ER (reticulophagy), and microbes (xenophagy), among others. In mammalian cells, three primary types of autophagy, depending on how the cargo is delivered to the lysosome—macroautophagy, microautophagy, and chaperone-mediated autophagy (CMA)—have been described [42]. The present review focuses on the macroautophagy process (hereafter referred to as autophagy) which takes place through the following four steps: (1) initiation, (2) nucleation, (3) elongation, and (4) fusion and degradation [43,44,45], as described in detail in Figure 2. 

The autophagic process is tightly regulated depending on nutrient status by different signaling pathways that include the mammalian target of rapamycin complex 1 (mTORC1), AMP-activated kinase (AMPK), and Sirtuin-1 (SIRT1) pathways. Autophagy is negatively regulated by the mTORC1 pathway, which also regulates several important and essential processes including cell growth and protein synthesis [46]. Under nutrient-rich conditions, mTORC1 is activated and phosphorylates ULK1, thereby preventing its activation by AMP activated protein kinase (AMPK), a key activator of autophagy [47]. Moreover, mTORC1 phosphorylates and inhibits the nuclear translocation of the transcription factor TFEB, which drives the expression of genes for lysosomal biogenesis and the autophagy machinery [46].

Conversely, mTORC1 is inhibited following nutrient starvation or energy deprivation and, thus, AMPK as a key energy sensor is activated and autophagy is induced [47]. When AMPK becomes activated, it phosphorylates and activates the tuberous sclerosis complex (TSC) which is composed of the TSC1 (hamartin), TSC2 (tuberin), and TBC1D7 proteins. The TSC complex is an essential inhibitor of mTORC1 activity through the activation of its GAP (GTPase-activating protein) activity towards the small G-protein Rheb (Ras homologue enriched in brain) [48]. Inactivation of mTORC1 activity by the TSC complex occurs via lysosomal recruitment of cytosolic TSC complex, where mTORC1 is located [49].

In addition, AMPK activation stimulates the activity of the nicotinamide adenine dinucleotide (NAD^+^)-dependent deacetylase SIRT1 by elevating the intracellular levels of its co-substrate, NAD^+^, and inducing lifespan extension [50]. Regarding post-translational modifications, the acetylation status of histones as well as cytoplasmic proteins is related with the regulation of autophagic flux [51,52]. In this sense, growing evidence indicates that both the activity of sirtuins, the deacetylation protein status, and the activation of autophagic flux decline with age [53].

Upon starvation, hypoxia, mitochondrial dysfunction, or several other circumstances in which a high intracellular levels of ROS/RSN are achieved, the c-Jun N-terminal kinase (JNK), or stress-activated protein kinase, becomes activated. JNK activation mediates the phosphorylation of B-cell lymphoma 2 (BCL2) protein to induce apoptosis. On the contrary, when JNK is not activated, dephosphorylated BCL2 interacts with beclin-1, which is a BCL2-interacting protein mediating the inhibition of beclin-1—dependent autophagy. This is only one of the main mechanisms of the complex crosstalk between autophagy and apoptosis [54].

Mitochondrial quality control mechanisms involved the activity of the phosphatase and tensin homolog (PTEN)-induced kinase 1 (PINK1) and an ubiquitin E3 ligase known as Parkin. PINK1 protein acts as a molecular sensor of damaged mitochondria and promotes the recruitment of Parkin on the outer mitochondrial membrane (OMM) resulting in the ubiquitination of numerous OMM proteins, which in turn recruits other proteins to mitochondria to initiate mitophagy. It is well known that alterations of this mitophagy and/or autophagy pathways lead to the accumulation of altered proteins and damaged organelles contributing to oxidative stress. 

In addition, another adaptive cellular response and a redox signaling axis that confers cellular protection against oxidative stress is the nuclear factor erythroid 2-related factor 2 (Nrf2)-Kelch-like ECH-associated protein 1 (Keap1)-antioxidant response elements (ARE) pathway. Activated Nrf2 is translocated to the nucleus where it activates the transcription of antioxidant genes, including heme oxygenase-1 (HO-1) [55]. Of note, and regarding the crosstalk between the antioxidant effects mediated by Nrf2-Keap1-ARE activation and autophagic process, it has been described that p62/SQSTM1 protein, which is degraded by autophagic process, also represents one of the Nrf2 target. Moreover, p62/SQSTM1 protein competitively binds to the redox sensor Keap1, disrupting its interaction with Nrf2, resulting in Nrf2 activation [56]. As a consequence, this p62/SQSTM1 positive-feedback loop may have beneficial effects against oxidative stress-dependent cell death through the activation of mitophagy by increasing the degradation of defective and altered mitochondria [38]. Other mechanisms are involved in autophagy regulation by oxidative stress such as post-translational modifications of key autophagy regulators. In this sense, recent studies demonstrate that some plant polyphenols including resveratrol, curcumin, and quercetin are able to activate the Nrf2-Keap1-ARE, counteracting oxidative damage and representing a novel therapeutic approach based on the antioxidant properties of these natural compounds [29,57,58].

Moreover, there has been a growing interest in the beneficial effects of the dietary consumption of polyphenols beyond its antioxidant activity, because most of them are able to activate autophagy and, thus, play a beneficial role in the redox balance by alleviating oxidative stress. Therefore, dietary polyphenols have emerged as a promising therapeutic strategy to prevent the development of several diseases including metabolic and neurodegenerative ones [57,59].

Resveratrol (2,3,4′-trihydroxystilbene), which is a caloric restriction mimetic (CRM), stimulates in vivo SIRT1, inducing the kinase activities of liver kinase B1 (LKB1) and AMPK, and inhibiting mTORC1 signaling, leading to autophagy activation [60]. In addition, resveratrol reduces ROS production in response to D-galactose of aging cardiomyocytes, enhancing cardiac function and diminishing ischemia/reperfusion-induced cell apoptosis through the regulation of mitophagy [61]. Moreover, resveratrol could enhance the association between mTORC1 and its inhibitory protein DEPTOR (DEP-domain containing mTOR-interacting protein), thus triggering the inactivation of mTORC1 [62]. It also increases in vivo mitochondrial biogenesis and function [60].

Flavonoids such as fisetin and quercetin lead to autophagy induction by TFEB nuclear translocation via mTORC1 inhibition [63,64]. The polyphenol curcumin is able to translocate TFEB to the nucleus, leading to autophagy activation by mTORC1-independent inhibition [65]. In addition, curcumin activates AMPK, which in turn phosphorylates BCL2, and subsequently disrupts the interaction between BCL2 and BECN1, resulting in autophagy activation [66] or upregulation of the expression of BECN1 protein which reduces p62 levels [67]. More recently, it has been demonstrated that the dietary tannin punicalagin activates autophagy through the inhibition of Akt signaling pathway, resulting in the activation of the transcription factor forkhead box O3a (FOXO3a) [68]. FOXO transcription factors have been identified to promote the expression of *ATG* genes, including *LC3b* (*Map1lc3b*), *Gabarapl1, Pi3kIII, Ulk2, Atg12l, Beclin1, Atg4b*, *Atg14*, and *Bnip3*, then leading to an increase in both autophagic and mitophagic flux [69].

Oleuropein and its related compounds, which are secoiridoids present in olive oil, have also emerged as autophagic inductors because they could activate the Ca^2+^/CaMKKβ/AMPK signaling pathway and hence preserve TFEB in its activated state [70] or even induce the expression of proteins involved in autophagic process including BECN1 and LC3 [71,72] mainly caused by AMPK activation [73]. Besides all these specific mechanisms, both mono- and polyphenols could also activate autophagic flux by stimulating the deacetylase activity of SIRT1 and, consequently, downregulating the general acetylation status of cytosolic proteins [74]. 

## 3. Dietary Polyphenols and Type 2 Diabetes, Obesity, and Metabolic Syndrome

The mechanism of action of different polyphenols in the control of several metabolic diseases, including obesity and type 2 diabetes mellitus (T2DM) in relation with autophagy modulation, has been extensively studied. It is well known that polyphenols are a group of molecules with positive effects in the modulation of aging, a period in which most of the reviewed diseases show a higher prevalence. T2DM is a very complex disease and is considered as epidemic worldwide. It is the resultant from multiple factors in a progressive manner, being insulin resistance and pancreatic β cell dysfunction the most relevant. T2DM is associated with increased levels of glucose and lipids that could contribute to β-cell death and is characterized by two different phases. During the first one, which main event is insulin resistance with normal levels of glycemia, pancreatic β-cells increase their mass by hyperplasia or hypertrophy, with a concomitant insulin and amylin secretion [75]. The duration of this phase depends on the patient’s idiosyncrasy and can be extended for years. At the final stage, pancreatic β-cells fail and then hypoinsulinemia appears. In addition, amylin deposition occurs in nearly 90% of T2DM patients. During the progression to T2DM, a chronic activation of mTORC1 signaling pathway has been detected. Although this activation is necessary during the first phase of the disease, for insulin resistance compensation by pancreatic β-cells, its chronic effect is deleterious for these cells [76,77]. Importantly, this maintained activation of mTORC1 generates a blockade in autophagic flux, which is essential for a correct elimination of damaged organelles or protein aggregates. These effects generate a continuous stimulation of the unfolded protein response (UPR), which in turn upregulates ER protein chaperones in charge of promoting protein folding. However, when the ER protein folding capacity is overwhelmed, cells undergo a condition of chronic ER stress, triggering the activation of apoptosis [78,79]. Nowadays, T2DM is considered a disease affecting the folding ability of pancreatic β-cells. In fact, the expression of different endogenous ER chaperones such as the 78-kDa glucose-regulated protein (GRP78) or BiP protein and protein disulfide isomerase, or chemical chaperones, such as taurine-conjugated derivative from ursodeoxycholic acid (TUDCA) or 4-phenyl butyric acid (4-PBA), diminished β cell failure and facilitated the correct folding, avoiding protein aggregation and apoptosis [80]. In this regard, azoramide, a small molecule which modulates UPR activity, exerts a strong antidiabetic activity [81,82]. In addition to all the changes mentioned before through the progression to T2DM, mitochondrial dysfunction is also shown. In general terms, mitochondrial dysfunction occurs as the natural history of aging [83]. However, during T2DM progression a concomitant defect in the mitochondrial clearance or mitophagy occurs which promotes an accumulation of aberrant and dysfunctional mitochondria which cannot be eliminated by mitophagy [84]. In this regard, a chronic activation of mTORC1 in pancreatic β-cells is able to inhibit both general autophagy as well as mitophagy, with a higher production of oxidized mitochondrial proteins and an accumulation of mitochondria with an altered membrane potential, which are not degraded by mitophagy [85]. One of the mechanisms that could explain this failure in mitophagy is the reduction in the levels of PINK1, which is an essential component in the recognition of mitochondria with an altered membrane potential as observed in fibroblasts with mTORC1 hyperactivation [86]. In addition to that, the overexpression of human amylin in pancreatic β cells impairs both bulk autophagy as well as mitophagy [87]. More importantly, these defects have been observed in prediabetic stages as well as in T2DM patients [88]. 

One of the most characterized and studied polyphenols with respect to its effects in diabetes is the stilbene compound resveratrol. Apart from its protective effects in diabetes, resveratrol can be a positive treatment in diabetes-related pathologies such as diabetic cardiomyopathy [89]. In fact, positive effects of resveratrol in the reduction of oxidative stress [90] as well as an increase in the sensitivity to insulin [91] with a concomitant decrease in lipid levels have been assessed [92]. In addition to these actions, resveratrol has been proposed to modulate the expression of both pro-apoptotic and anti-apoptotic factors [31]. Furthermore, stilbenes can stimulate an important antioxidant defense such as the transcription factor called Nrf2. Nrf2 is involved in the control of the transcription of different antioxidants in response to inflammation and oxidative stress [93]. Stilbenes are also involved in the activation of autophagy, through the modulation of p62 protein levels and SIRT1 activity [94]. An essential role of SIRT1 has been proposed for the induction of autophagy in response to hypoxia in a T2DM rat model [95]. Resveratrol stimulates autophagic flux, improving diabetic cardiomyopathy in a diabetic mouse model mediated by a decrease in p62 protein levels, facilitating SIRT1 activity. Moreover, SIRT1/FOXO/Rab7 has been proposed as a potential therapeutic pathway which mediates the effects of resveratrol on autophagic flux and ameliorates dysfunctional autophagy in diabetic cardiomyopathy [96]. Resveratrol can also modulate autophagy by alternative indirect mechanisms which involve transcriptional regulation of microRNA’s (miRNA’s for short). miRNA-18a-5p is one kind of miRNA which expression is enhanced in a diabetic mouse model after treatment with resveratrol [97].

Other polyphenols with a wide protective effect in diabetes are curcumin and its analogues. These groups of molecules are able to down-regulate the chronic stimulation of ER stress, reducing apoptosis [98] and increasing insulin sensitivity [99]. There are multiple evidences indicating a pro-autophagic role of curcumin in order to moderate the negative effects of T2DM [66]. For instance, curcumin can modulate several proteins involved in autophagy including LC3, p62, and beclin-1, among others. These effects have been described in a diabetic mice model treated with curcumin, preventing podocyte apoptosis during diabetic nephropathy [67].

(−)-Epigallocatechin-3-gallate (EGCG) is a polyphenol isolated from green tea with a significant effect in autophagy modulation. EGCG is able to regulate autophagy at different levels, alleviating part of the deleterious effects observed in diabetic patients. For instance, it can avoid lipid accumulation in vascular endothelial cells, increasing the degradation of lipids by lipophagy, and facilitating the recognition between LC3B (located in the autophagosomes) and the lipid droplets for their catabolism [100]. This polyphenol has a protective effect for both mitochondrial dysfunction and aberrant autophagy described in the heart’s tissue of diabetic rats [101,102].

Punicalagin, one of the main polyphenolic components of pomegranate, is able to protect liver dysregulation in response to T2DM. In fact, this molecule has been proposed very recently as an autophagy inducer through the modulation of different protein markers such as LC3B and p62 which allow the reactivating of autophagy [68]. Furthermore, oleuropein has been related to an induction of autophagy as part of its mechanism of action [73]. Oleuropein, together with hydroxytyrosol, is involved in the induction of autophagy through the regulation of AMPK/mTORC1 signaling pathway [70,103].

Obesity, one of the main factors associated to the development of T2DM, has been implicated in autophagy dysregulation. In fact, a dysregulation in autophagic activity plays an important role in the appearance of insulin resistance [104]. Furthermore, adipocyte hypertrophy occurring in obesity contributes to insulin resistance as well, by the production of different inflammatory cytokines which lead to an increased in endoplasmic reticulum stress (ER-stress) [105]. One of the main polyphenols with protective properties in obesity and, specifically in liver steatosis, is resveratrol. In rats, resveratrol effect is similar to an energy restriction (a reduction of 15% of calories in energy intake), with a decrease in p62 protein levels and an increase in LC3B, beclin-1, and atg5 protein levels, which collectively suggests an autophagic activation [106]. Apart from resveratrol, other polyphenolic compounds with protective effects through autophagy activation in obesity have been assessed. A treatment with epigallocatechin-3-gallate (EGCG) for two weeks in a mice model of high-fat diet (HFD)-induced obesity stimulated both autophagic flux in white adipose tissue in an AMPK-dependent manner and beclin-1 activation [107]. Very interestingly, the effect of a 30 days’ supplementation with resveratrol on gene expression and adipose tissue morphology in a cohort of obese men has been reported. The authors indicated that, at the end of the study, there was a significant reduction in adipocyte size, with a downregulation of both Wnt and Notch signaling pathways. In addition, there was an upregulation in different proteins involved in the cell cycle control which suggested an increase in adipogenesis. Finally, an increase in lipophagy accompanied by a reduction in inflammation were also observed [108].

The metabolic syndrome represents several injuries including obesity, T2DM, and nonalcoholic fatty liver (NAFL); a mitochondrial dysfunction appears, thus contributing to insulin resistance. Then, mitophagy could potentially preserve mitochondrial function by the elimination of these damaged organelles [109,110]. Another important consequence of obesity is the appearance of NAFLD. It is well known that the elimination of lipid accumulation that occurs in the liver by the activation of lipophagy contributes to a decrease in NAFLD pathology [111]. In this regard, there are multiple examples assessing the protective effect of different polyphenols: pro-autophagic activity of resveratrol has been demonstrated both in vitro model of palmitic acid-induced hepatic steatosis [112] and in in vivo models of NAFLD, where it also reduces liver inflammation [113,114]. In fact, this in vivo beneficial effect of resveratrol was associated with autophagy activation, because the elimination of ULK1, which is essential in the control of autophagosome formation, impaired the protective role of resveratrol [113]. Using a cafeteria diet-induced obesity in rats, polyphenols from bergamot were able to induce lipophagy and prevent NAFLD onset. In the bergamot-treated animals, a reduction in p62 protein levels and an increase in both LC3B and beclin-1 protein levels were shown, both events being associated with an activation of the autophagic flux. All these effects were linked to a decrease in both liver lipid accumulation and inflammation [115,116]. Furthermore, an in vitro model of liver cells (HepG2) treated with different levels of fatty acids (a combination of oleic acid and palmitic acid) which are found in patients with liver steatosis, assessed the accumulation of lipids inside the cells. Very interestingly, the protective effect of different polyphenols derived from *Toona sinensis* A. Juss. was associated with an activation of AMPK/ULK1 signaling pathway with a concomitant increase in LC3B protein levels [117]. Similar protective results related to lipid accumulation in a mice model of NAFLD in response to blueberry polyphenols were reported [118]. Very recently, the protective effect exerted by a polyphenol extract from apple in the activation of lipophagy and restoration of lysosomal pH was related to intervention through the SIRT1/AMPK signaling [119]. In fact, polyphenols such as quercetin can stimulate lysosomal biogenesis and, hence, the clearance capacity of the cells by facilitating the nuclear translocation of TFEB [64]. Although these effects have been observed in retinal epithelium cells (RPE cells), it should be of great interest to explore the improvement of the TFEB-lysosomal axis in other pathologies such as T2DM, obesity, and metabolic syndrome. In summary, autophagy activation represents a key event in the pathophysiology of fatty liver disease and its modulation by the use of different polyphenols could contribute to prevent both the disease and related pathologies such as T2DM, obesity, and other metabolic complications [120].

## 4. Effect of Polyphenols on Cardiovascular Diseases

Cardiovascular diseases are a direct consequence of aging. During the aging process, there is a progressive deterioration of mitochondrial function and a decline in cardiac efficiency. Using a cellular model of senescent-like cardiomyocytes, treatment with resveratrol improved cardiac function by the activation of SIRT1 signaling pathway. In addition, the effect of resveratrol in the correct function of two essential proteins in mitophagy namely PINK1 and PARKIN was also assessed [61]. A polyphenol mixture from green tea (catechin > 90% and EGCG > 70%) exerted a protective role in the reduction of atherogenesis in an ApoE-knock-out mice model through the activation of autophagy. In this model, there was an increased production of p62 protein levels linked to an increased autophagosome formation flux, with an increase in LC3B and beclin-1 protein levels in the polyphenol-treated group [121]. Oleuropein aglycone has been studied in a cellular model with an increased oxidative stress in cardiomyocytes. In these cells, there was an induction of autophagy, detected by an increase in LC3B and beclin-1 protein levels. However, very importantly, there was an improvement of TFEB translocation to the nucleus which suggested an improved autophagy induction [72]. In this regard, different polyphenols have been involved in the protection against cardiovascular disease and vascular inflammation, including EGCG, quercetin, resveratrol, apigenin, and curcumin, through autophagy modulation among other mechanisms [122,123]. Recent reports indicate that resveratrol, by autophagy activation could be useful for the treatment of the ischemic process found in diabetic myocardium. The beneficial effect of resveratrol was associated with an increase in beclin-1 and LC3B protein levels and a reduction in inflammation, by diminishing two inflammatory cytokines such as TNF-alpha and IL-6 [124]. In general terms, mTOR signaling pathway has been proposed as a target for treating atherosclerosis, cardiac hypertrophy, and heart failure. Then, every natural compound with proven activity towards the inhibition of mTORC1, could represent an interesting approach to reduce both the above cited diseases and the number of deaths caused by cardiovascular disease worldwide [125].

## 5. Dietary Polyphenols and Neurodegeneration

Dysfunction of autophagy contributes to protein misfolding and aggregation which underly the pathogenesis of human proteinopathies such as Alzheimer (AD), Parkinson (PD), and Huntington diseases [126]. Protein homeostasis is essential for the cell function but it is especially relevant in postmitotic neurons, which have a limited regenerative potential and thus, mechanisms involved in the elimination of damaged organelles or protein aggregates are crucial for preserving cell survival [127]. Although the mechanisms that control the above cited diseases are different, they share a common feature which is the accumulation of different protein aggregates in neurons, leading to cell toxicity and neurodegeneration. As mentioned for diabetes, neurodegenerative diseases prevalence increases during aging. It is well known that all the systems involved in protein quality control including UPS and autophagy-lysosome ones fail with aging, thus promoting the accumulation of aggregates which are not present in younger organisms. 

In the case of Alzheimer’s patients, there is an accumulation of amyloid β (Aβ) and an intracellular accumulation of an hyperphosphorylated form of Tau protein which leads to a structure called neurofibrillary tangles. Although a small percentage of patients (1%) with Alzheimer’s disease shows a mutation in one of the genes involved in the disease, including presenilins 1 and 2 or amyloid precursor protein (APP), most patients (around 90%) are called sporadic ones and depend on both genetic and environmental determinants [128]. Aβ oligomers and the hyperphosphorylated form of tau produce several alterations in neurons including inhibition of proteasome and autophagy-lysosome degradation systems, which contribute to a higher accumulation of these proteins [129]. In addition, it has been shown that Aβ is able to accumulate lysosomes inside itself, thus altering their membrane permeability [130]. Tau protein is involved in both microtubule assembly and stability, depending on its phosphorylation degree, which is the result of the activity of kinases as well as phosphatases. Microtubule stability is associated with an unphosphorylated form of tau protein [131,132]. In contrast, hyperphosphorylation of tau provokes a microtubule instability which generates an alteration in the polymerization of microtubules, altering the correct transport and movement of different organelles, which are in turn essential processes in the regulation of synapsis [133].

With respect to PD, another protein known as α-synuclein is involved. This protein can assemble into aggregates and form new deposits known as Lewy bodies (LB) mainly in dopaminergic neurons [134]. In the same line with AD, different mutations related to the increased capacity of this protein to aggregate have been described for PD. This aggregation alters normal cellular physiology and organelles, including ER as well as mitochondria, which finally contribute to cell death [135,136]. Multiple genes exist associated with the disease such as *LRRK2, SNCA, LAMP3*, and *ATP13A2*. Although *SNCA*, which encodes α-synuclein, is the major genetic alteration related with the pathology, other genetic mutations are also involved in the mechanisms related to the control of the mitochondrial quality, such as PINK1 and protein deglycase (DJ-1). However, other different mechanisms have been found to be altered as a consequence of α-synuclein aggregation, including vesicular trafficking between compartments such as ER–Golgi transport and Golgi apparatus fragmentation [137,138]. In addition, several posttranslational modifications such as phosphorylation, ubiquitination, and acetylation have been described to alter the structure of α-synuclein, its correct function, and its aggregation capacity [139]. Very interestingly, an hyperphosphorylation state of α-synuclein has been related with a change in its subcellular localization, leading to an interaction with histones and the inhibition of their activity inducing neurotoxicity and cellular apoptosis [140,141,142]. This phosphorylation could be mediated by different kinases in cells. Although the exact mechanism remains unknown, polo-like kinases (PLK) could be responsible for regulating this phosphorylation event in the serine 129 [143]. Apart from phosphorylation, lysine acetylation status of α-synuclein has been involved in the pathogenesis of the disease as well [144]. In addition, this protein modification impairs mitophagy in fibroblasts, as shown for PD patients [145].

Autophagy modulation is a key avenue for the treatment of neurodegenerative diseases as it has been very recently reviewed [146]. In this regard, different polyphenols have been proposed as protective in different neurodegenerative diseases such as AD and PD. Curcumin is able to modulate autophagic machinery in order to restore the defects observed in the neurons of patients affected with these diseases [59], and it is also able to directly interact with α-synuclein, then avoiding its aggregation and hence its toxicity [147]. This effect on aggregation which reduces synaptic toxicity in response has been described not only for curcumin, but also for other phenolic compounds such as myricetin, rosmarinic acid, norhydroguaiaretic acid, and ferulic acid [148]. A synthetic form of curcumin, called C1, is able to activate autophagy by inducing TFEB, a master transcription factor for lysosomal biogenesis as well as autophagy machinery [65]. Using a fly model of PD associated with a loss of function of an essential protein involved in mitophagy, a grape skin extracts exerted a protective effect by promoting autophagy and preserving mitochondrial function, provoking a rescue in mitochondrial structural defects by the activation of mitophagy [149]. In this regard, alpha-arbutin obtained from different species of the Ericaceae family, protects from rotenone-induced mitochondrial dysfunction in a neuroblastoma cell line (SH-SY5Y). The protective effect of this polyphenol is due to its ability to induce both AMPK and autophagy [150]. Similarly, resveratrol protects mitochondrial function, by enhancing autophagic flux, in a mitogen-activated protein kinase (MEK)-dependent manner, indicating that this signaling pathway is essential in the actions of resveratrol, at least in the modulation of mitochondrial dynamics [151]. Very recently, a polyphenol-enriched extract from lychee seed has demonstrated a neuroprotective effect through the activation of autophagy by increasing beclin-1 and LC3B protein levels, mainly by AMPK activation [152].

Another mode of action of polyphenols is by acting on stem cells. In general terms, neuroprotection by polyphenols and stimulation of neurogenesis [153] occurs at different levels, including modulation of antiapoptotic proteins, stimulation of different signaling pathways inside the cells, inhibition of pro-oxidant enzymes, and alteration of mitochondrial function. Furthermore, polyphenols exert an active role in quenching free radical species and chelating metal ions, with the ability to regulate the major degradative pathways of proteins [154]. Oleuropein is involved in alleviating oxidative stress as well as inducing autophagic flux [155]. In this regard, it is also involved in the profound reduction of β-amyloid deposits, with a concomitant increase in autophagy activation. In parallel, there is a great reduction in mTORC1 signaling pathway in response to oleuropein [71]. Oleuropein aglycone induces autophagy and favors the elimination of protein aggregates, decreasing the cognitive decline [156]. Using a mice model of AD, the treatment with hydroxytyrosol increases autophagic flux mainly by activation of MAPK signaling pathway this resulting in neuroprotection [157]. Resveratrol and other compounds called sirtuin activating compounds or STACS have been assessed as strongly inducing SIRT1 [158]. SIRT1 inhibits c-Jun by deacetylation, decreasing the transcriptional activity of the activator protein-1 (AP1) involved in the expression of several inflammatory and stress genes [159,160]. The essential role of SIRT1 as autophagy regulator derives from the fact that SIRT1 is able to bind to an ATG protein, regulating its activity by deacetylation and, hence, autophagy [161]. 

Figure 3 depicts the main molecular targets of some polyphenols and their beneficial effects on metabolic, cardiovascular, and neurodegenerative diseases. 

**Table 1 antioxidants-10-00142-t001:** Summarizes some of the polyphenols-mediated actions on autophagy and other important biological effects.

Table 1. Examples of Health Benefits of Dietary Polyphenols by Autophagy Activation					
Class of Polyphenol	Name of Polyphenolic Compound	Molecular Formula	Molecular Weight (g/mol)	Biological Effects	References
Stilbenes	Resveratrol 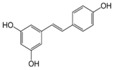	C_14_H_12_O_3_	228.25	Improves mitochondrial function in vitro and in vivo	[60,149,151]
				Autophagic activation in vitro and in vivo	[52,61,94,95,96,97,106,113,122,124,149,151,158]
				Improves insulin sensitivity in humans	[94]
Flavonoids	Fisetin 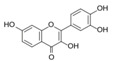	C_15_H_10_O_6_	286.24	Autophagic activation in vitro	[63]
	Quercetin 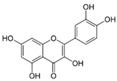	C_15_H_10_O_7_	302.23	Autophagic activation in vitro and in vivo	[64,122,162,163,164,165,166,167,168]
	Myricetin 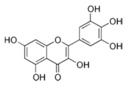	C_15_H_10_O_8_	318.23	Inhibition of α-synuclein aggregation in vitro	[148]
	Apigenin 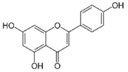	C_15_H_10_O_5_	270.24	Autophagic activation in vitro	[122,169,170,171,172]
	Luteolin 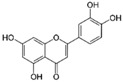	C_15_H_10_O_6_	286.24	Autophagic activation in vitro and in vivo	[173,174,175,176,177,178]
	Baicalein 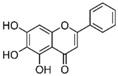	C_15_H_10_O_5_	270.24	Autophagic activation in vitro and in vivo	[179,180,181,182,183]
Phenolic acids	Rosmarinic acid 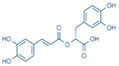	C_18_H_16_O_8_	360.31	Inhibition of α-synuclein aggregation in vitro	[148]
	Nordihydro-guaiaretic acid 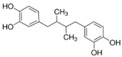	C_18_H_22_O_4_	302.4		
	Ferulic acid 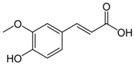	C_10_H_10_O_4_	194.18		
Tannins	Punicalagin 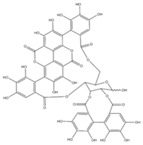	C_48_H_28_O_30_	1084.71	Autophagic activation in vitro and in vivo	[68]
Secoiridoid	Oleuropein 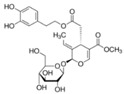	C_25_H_32_O_13_	540.51	Autophagic activation in vitro and in vivo	[70,71,72,73,155,156]
Catechins	(-)-epigallocatechin-3-gallate 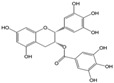	C_22_H_18_O_10_	442.4	Autophagic activation in vivo	[101,102,107,121]
Polyphenols	Curcumin 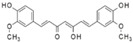	C_21_H_20_O_6_	368.38	Autophagy induction in vitro and in vivo	[59,65,66,67,123]
				Downregulation of ER stress	Reviewed in [98]
				Increase insulin sensitivity in vitro and in vivo	[99]
				Inhibition of α-synuclein aggregation in vitro	[147]

## 6. Conclusions

Natural products with dietary polyphenols among them have emerged as a promising group of potential agents in health promotion, supported by their structure and results from molecular studies. Polyphenols may act on multiple molecular targets in relation to autophagic flux activation with potential beneficial effects on metabolic and neurodegenerative diseases. In this review, different protein targets and signaling pathways modulated by these natural products have been discussed.

Since years, the positive effects of a variety of plant extracts and dietary polyphenols including resveratrol, curcumin, epigallocatechin-3-gallate, punicalagin, oleuropein, myricetin and rosmarinic, norhydroguaiaretic, and ferulic acids have been reported. These compounds are CRM and anti-aging molecules which may activate AMPK and SIRT1 activities by increasing intracellular levels of AMP and NAD^+^, leading to mitochondrial function improvement and autophagy stimulation. Activation of autophagy removes protein aggregates, lipid droplets, and damaged organelles, stimulates antioxidant defenses, and alleviates both ER and oxidative stress mediated by mTORC1 hyperactivation, resulting in an enhancement of cell survival. 

In conclusion, in view of their potential beneficial effects, polyphenols represent a promising chemical group for which further studies, including clinical trials, should be conducted for their in-depth knowledge which would likely be helpful for translational outcomes.

## Figures and Tables

**Figure 1 antioxidants-10-00142-f001:**
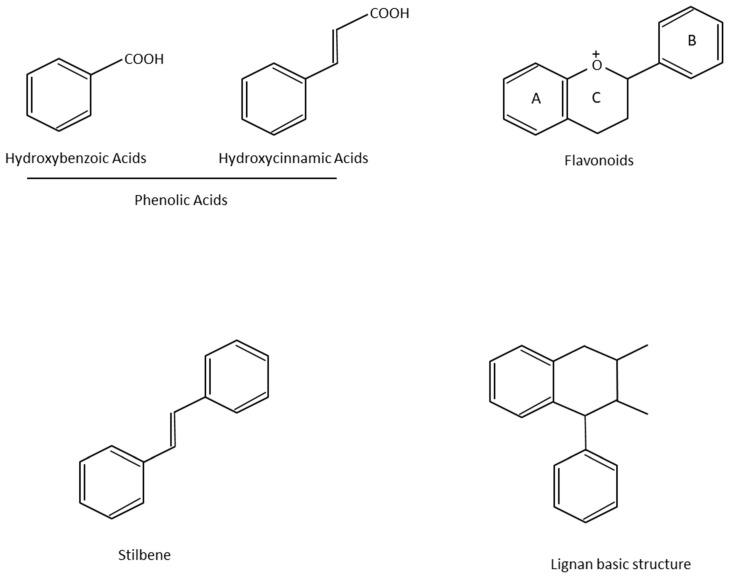
Basic chemical structures of the main types of polyphenols with biological interest.

**Figure 2 antioxidants-10-00142-f002:**
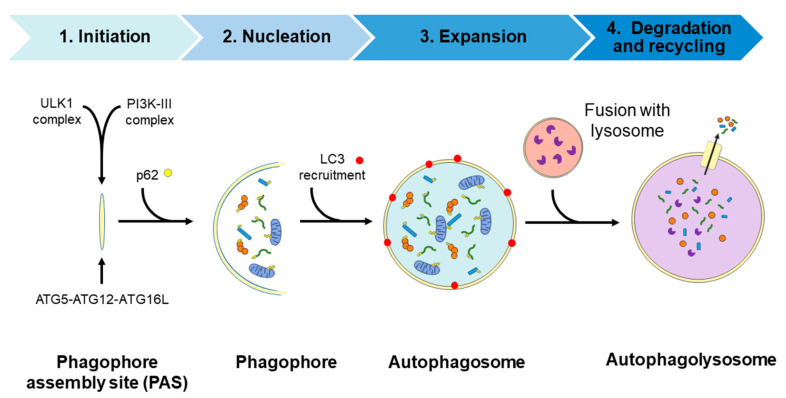
Diagram of the autophagic process. Steps: (1) Initiation, with the formation of the phagophore assembly site (PAS) which assemblies the Atg1/ULK1 complex and the PI3K-III complex; (2) nucleation step occurs upon the recruitment of the ubiquitin-like conjugation system (the ATG5/ATG12/ATG16L complex) to the PAS. Then, the curvature of this double-membrane (phagophore) is induced to facilitate the sequestration of the cargo. The recruitment of the ubiquitinated proteins or damaged organelles to the phagophore is carried out by the p62 protein, which links the autophagy pathway and the ubiquitin-proteasome system (UPS) by binding the ubiquitinated proteins to LC3 protein for autophagic degradation; (3) the elongation of the phagophore to form a spherical vesicle termed autophagosome; (4) finally, autophagosomes fuse with lysosomes to form autophagolysosomes/autolysosomes and the autophagic cargo are degraded by the action of resident lysosomal/vacuolar hydrolases and then are exported to the cytosol for reuse by the cell. ATG, autophagy-related; LC3, microtubule-associated protein 1A/1B-light chain 3; p62, sequestosome 1 (SQSTM1); PI3K-III, class III phosphatidylinositol 3-kinase; ULK1, unc-51-like kinase.

**Figure 3 antioxidants-10-00142-f003:**
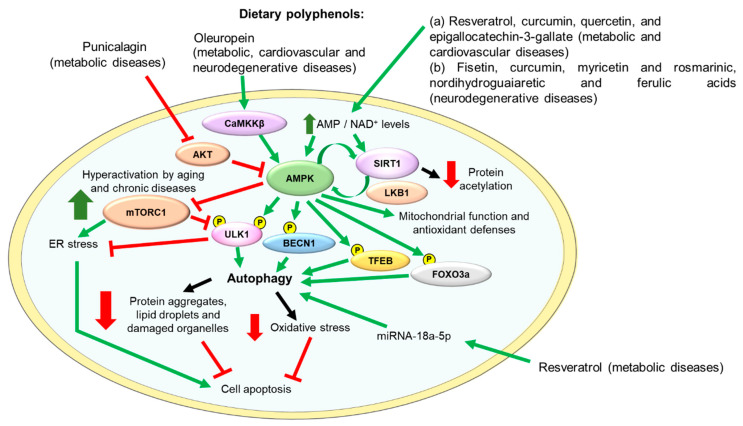
Main molecular mechanisms involved in the antioxidant activity and the beneficial effects of dietary polyphenols in metabolic, cardiovascular, and neurodegenerative diseases. The dietary polyphenols mentioned in this figure have been demonstrated to have a beneficial impact on the prevention of these diseases by different mechanisms including increasing the intracellular levels of AMP and NAD^+^, activating CaMKKβ or inhibiting AKT. The regulation of these molecular pathways results in the activation of both AMPK and SIRT1 activities. AMPK activates SIRT1, and SIRT1 activates AMPK by inducing LKB1 activity. Activated AMPK diminishes mTORC1 pathway activation, reducing ER stress, activates autophagy, and stimulates mitochondrial function and antioxidant defenses, and, consequently, reduces oxidative stress and cell apoptosis. Green arrows indicate activation and red lines indicate inhibition of the activity of the target protein. AMPK, adenosine monophosphate (AMP)-activated protein kinase; AKT, protein kinase B; BECN1, beclin-1; CaMKKβ, calcium/calmodulin-dependent protein kinase kinase β; ER, endoplasmic reticulum; FOXO3a, forkhead box O3a; LKB1, liver kinase B1; miRNA, microRNA; mTORC1, mammalian/mechanistic target of rapamycin complex 1; NAD+, nicotinamide adenine dinucleotide; Nrf2, Nuclear factor (erythroid-derived 2)-like 2; SIRT1, sirtuin-1; TFEB, transcription factor EB; ULK1, unc-51-like kinase 1.

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
