# Peer review of "Dietary Polyphenols in Metabolic and Neurodegenerative Diseases: Molecular Targets in Autophagy and Biological Effects"

_antioxidants, 2021, doi:10.3390/antiox10020142_

Round 1

Reviewer 1 Report

Dear Authors of

Dietary polyphenols in metabolic diseases and neurodegeneration: Molecular targets on autophagy and biological effects

Manuscript ID antioxidants-1002268   The proposed topic is quite interesting and innovative However, I find some points that need probable minor revision interventions.   -Legends of Figures need to be expanded, resuming the descrived processes and molecules (protein, enzymes, ect9 involved.    Also, polyphenoles are poorly described in their biochemical and biophysical properties. For example, molecular weights, ect........ should be reported,    -Figure 2: change Autolysosome into Autophagolysosome   lane 450 Acknowlegdments are not completed and leaved in an uncommon form   Minor points lane 180: Figure 2 instead of figure 2. make uniform citation style e.g. lane 143 (13,14) vs lane 325 (57); (58), ect   lane 352  Aoligomers  ? lane 355 Acan ?    

Author response

Reviewer 1

English language and style

( ) Extensive editing of English language and style required
( ) Moderate English changes required
( ) English language and style are fine/minor spell check required
(x) I don't feel qualified to judge about the English language and style

Dear Authors of Dietary polyphenols in metabolic diseases and neurodegeneration: Molecular targets on autophagy and biological effects

Manuscript ID antioxidants-1002268  

The proposed topic is quite interesting and innovative However, I find some points that need probable minor revision interventions.  

- Thanks a lot to the reviewer for her/his commentaries about our review. We are profoundly grateful for these comments. We will be answer point-by point to all the suggestions highlighted by this referee.

Legends of Figures need to be expanded, resuming the described processes and molecules (protein, enzymes, ect9 involved. 

- Thanks a lot for the comment. We have expanded the figure legends of figure 2 and 3 in order to include all the described molecules involved

  Also, polyphenoles are poorly described in their biochemical and biophysical properties. For example, molecular weights, ect........ should be reported,   

- In order to resume all the main properties and its biological effects of dietary polyphenols, we have included in this revised version the new table 1 in this revised version of the manuscript, which includes the class of polyphenol, the name, molecular formula, molecular weight, biological effects and the corresponding references in the manuscript. We thank reviewer 1 for the suggestion of this change and with the inclusion of this table in the revised version, which greatly improve the quality of the work

Figure 2: change Autolysosome into Autophagolysosome  

-We have done this change (and also in the manuscript).

lane 450 Acknowlegdments are not completed and leaved in an uncommon form  

- We have removed acknowledgements section

Minor points

lane 180: Figure 2 instead of figure 2.

- We have done

make uniform citation style e.g. lane 143 (13,14) vs lane 325 (57); (58), ect  

- We have uniformed all the citation style

lane 352  Aoligomers  ?

- We have corrected

lane 355 Acan ?    

- We have corrected

Reviewer 2 Report

The review by Garcia-Aguilar et al. is written with a great clarity and simplicity, which is generally appreciated and welcome in scientific publications. English language is good and only minor style and spell errors have been detected. However, the introduction is too basic and occasionally reads like a PhD thesis, or a glossary. The sections 3 and 4 give e nice generic overview of type 2 diabetes mellitus (T2DM) and neurodegenerative diseases, but both sections are not complete. The referencing is also limited and often absent. These and other issues that should be addressed before resubmitting the review are reported below:

Major issues:

  • 1. The introduction section introduces basic terms such as redox balance, oxidative stress, signal transduction, apoptosis, polyphenols and so on, but does not relate one term to the other. While it can be occasionally done on purpose if totally unrelated topics, here it seems like a glossary. The purpose of an introduction is to give a short overview of the background of main topics (here autophagy and polyphenols) and then explain why these topics are related and why it is important to discuss them together.
  • 2. The introduction does not give a sufficient attention to "autophagy" as the main topic and does not explain why it is important to discuss the role of polyphenols in autophagy regulation. Finally, the authors give the background to oxidative stress and antioxidant effects of polyphenols, but they do not explain if and why antioxidant effects are linked to autophagy.
  • 3. The title of the review and the abstract are not consistent with the content of the section 3 which is limited to type 2 diabetes (T2DM) and focused on beta-pancreatic cells mainly. The authors should describe the role of different polyphenols in stimulation of autophagy in other metabolic diseases, related to metabolic syndrome, such as obesity, endothelial dysfunction, hypertension and dyslipidemia and cardiovascular complications. They should address non-alcoholic fatty liver disease or at least limitedly to its impact on T2DM. In fact, the hepatic autophagy plays an important role in contrasting T2DM. Beside that, there is a vast literature on effects of natural polyphenol extracts and mixtures on metabolic syndrome. Therefore, some relevant examples should also be mentioned in this section.
  • 4. The section 2 is not original at all since it is reporting only some basic mechanisms of autophagy without even mentioning polyphenols. Some examples of the impact of polyphenols on autophagy machinery should be mentioned and the authors should briefly discuss what is the crosstalk of oxidative stress and redox balance regulation on autophagy machinery in the context of polyphenols. Thus, the final title of this section should be: “Molecular mechanisms of autophagic process and its regulation by polyphenols”
  • 5. In the section 4, there are only two experimental papers with the examples of polyphenol-mediated effects on autophagy after the year 2016. The section should be updated, as many other papers have been published on this topic between 2017-2020.
  • 6. Both section 3 and 4 should be amended with the tables reporting the experimental papers related to polyphenol-mediated effects on autophagy
  • 7. When the authors prepare the new version of the review, they should pay more attention to references. In the present version, in sections 1 and 2 majority of sentences is without references.

Minor issues:

  • 8. The authors should be aware that the editorial office will screen for any examples of plagiarism in the final text of the review.
  • 9. The authors should present a more complete classification of polyphenols. For example, the sub-class of curcuminoids and other natural polyphenols are missing, while curcumin is often been mentioned as a polyphenol active on autophagy.
  • 10. lines 241-243 “The mechanism of action of different polyphenols in the control of several metabolic diseases, including obesity and type 2 diabetes mellitus (T2DM) in relation with autophagy modulation has been reported”. This sentence is not appropriate to start the chapter and it does not make a lot of sense. Maybe the authors intended …” has been extensively studied by several groups.” “Or has been extensively studied in recent years”
  • 11. 335-36 “Protein homeostasis is essential for maintenance in all the cells …..” This sentence is not clear.

Author Response

Reviewer 2

English language and style

( ) Extensive editing of English language and style required
( ) Moderate English changes required
(x) English language and style are fine/minor spell check required
( ) I don't feel qualified to judge about the English language and style

The review by Garcia-Aguilar et al. is written with a great clarity and simplicity, which is generally appreciated and welcome in scientific publications. English language is good and only minor style and spell errors have been detected. However, the introduction is too basic and occasionally reads like a PhD thesis, or a glossary. The sections 3 and 4 give e nice generic overview of type 2 diabetes mellitus (T2DM) and neurodegenerative diseases, but both sections are not complete. The referencing is also limited and often absent.

- Thanks a lot to the reviewer for her/his comments about our manuscript. All the authors hope that the answers given to this reviewer #2 will be helpful and useful for a better understanding of the manuscript

These and other issues that should be addressed before resubmitting the review are reported below:

Major issues:

  1. The introduction section introduces basic terms such as redox balance, oxidative stress, signal transduction, apoptosis, polyphenols and so on, but does not relate one term to the other. While it can be occasionally done on purpose if totally unrelated topics, here it seems like a glossary. The purpose of an introduction is to give a short overview of the background of main topics (here autophagy and polyphenols) and then explain why these topics are related and why it is important to discuss them together.

- We have rewritten the introduction section, including the terms suggested by the referee. We hope that this new introduction in the revised version will be a real overview of the background of the main topic.

  1. The introduction does not give a sufficient attention to "autophagy" as the main topic and does not explain why it is important to discuss the role of polyphenols in autophagy regulation. Finally, the authors give the background to oxidative stress and antioxidant effects of polyphenols, but they do not explain if and why antioxidant effects are linked to autophagy.

- Thanks a lot to the reviewer for this comment. We are completely agreed with this comment. As we mentioned before, we have rewritten the introduction in order to clarify all the basic interactions between the different processes. Then, in one side, we have shown the link between polyphenols and autophagy and, in addition, we have included the relationship between the antioxidant effect of polyphenols with autophagy as suggested by the referee.

  1. The title of the review and the abstract are not consistent with the content of the section 3 which is limited to type 2 diabetes (T2DM) and focused on beta-pancreatic cells mainly. The authors should describe the role of different polyphenols in stimulation of autophagy in other metabolic diseases, related to metabolic syndrome, such as obesity, endothelial dysfunction, hypertension and dyslipidemia and cardiovascular complications. They should address non-alcoholic fatty liver disease or at least limitedly to its impact on T2DM. In fact, the hepatic autophagy plays an important role in contrasting T2DM. Beside that, there is a vast literature on effects of natural polyphenol extracts and mixtures on metabolic syndrome. Therefore, some relevant examples should also be mentioned in this section.

- Thanks a lot to the referee for this comment. We are completely agreed with her/his comment and we think we have improved the quality of the manuscript. In order to include the effect of polyphenols in other metabolic diseases, we have changed the title and we have included new information in the 3rd section of the revised version, including obesity, non-alcoholic fatty liver disease and metabolic syndrome. Furthermore, we have included in this revised version of the manuscript, an additional section (section 4th of the revised manuscript), which describes the effects of polyphenols in cardiovascular complications (page 16).

  1. The section 2 is not original at all since it is reporting only some basic mechanisms of autophagy without even mentioning polyphenols. Some examples of the impact of polyphenols on autophagy machinery should be mentioned and the authors should briefly discuss what is the crosstalk of oxidative stress and redox balance regulation on autophagy machinery in the context of polyphenols. Thus, the final title of this section should be: “Molecular mechanisms of autophagic process and its regulation by polyphenols”

- Thanks a lot for this comment to the reviewer. First of all, we have changed the title of this section, as suggested by the reviewer. Furthermore, we have rewritten this second section of the manuscript, in order to give a wide overview of the regulation of autophagic machinery and oxidative stress by polyphenols.

  1. In the section 4, there are only two experimental papers with the examples of polyphenol-mediated effects on autophagy after the year 2016. The section should be updated, as many other papers have been published on this topic between 2017-2020.

- Thanks a lot to this reviewer for this comment. In addition to the changes explained before, in relation to the 5th section of the chapter (Dietary polyphenols and neurodegeneration), we have included in the revised version of the review, other experimental recent papers studying the effect of polyphenols in neurodegenerative diseases, as suggested by the referee.

  1. Both section 3 and 4 should be amended with the tables reporting the experimental papers related to polyphenol-mediated effects on autophagy

- Thanks a lot for this suggestion. We have included a new table 1, which includes, apart from the experimental papers related to the polyphenol-mediated effects on autophagy, suggested by this referee, the class of polyphenol, the name, molecular formula, molecular weight and their biological effects.

  1. When the authors prepare the new version of the review, they should pay more attention to references. In the present version, in sections 1 and 2 majority of sentences is without references.

- Thanks a lot for this suggestion and we are sorry for any mistake generated in the original version of the manuscript. We have included several references in section 1 and 2 and we have paid attention to the references before sending the final version of our review

Minor issues:

  1. The authors should be aware that the editorial office will screen for any examples of plagiarism in the final text of the review.

- Thanks a lot to the referee for this comment.

  1. The authors should present a more complete classification of polyphenols. For example, the sub-class of curcuminoids and other natural polyphenols are missing, while curcumin is often been mentioned as a polyphenol active on autophagy.

- Thanks a lot for the comment. In order to resume all the main properties and its biological effects of the dietary polyphenols mentioned in the revised manuscript, we have included the new table 1, which includes the class of polyphenol, the name, molecular formula, molecular weight, biological effects and the corresponding references in the manuscript. We thank reviewer 2 for her/his suggestion, which greatly improve the quality of the work

  1. lines 241-243 “The mechanism of action of different polyphenols in the control of several metabolic diseases, including obesity and type 2 diabetes mellitus (T2DM) in relation with autophagy modulation has been reported”. This sentence is not appropriate to start the chapter and it does not make a lot of sense. Maybe the authors intended …” has been extensively studied by several groups.” “Or has been extensively studied in recent years”

- Thanks for the comment and we are sorry for the mistake. We have corrected in the new version of the manuscript

  1. 335-36 “Protein homeostasis is essential for maintenance in all the cells …..” This sentence is not clear.

- Thanks for the comment and we are sorry for the mistake. We have corrected in the revised version of the manuscript

Reviewer 3 Report

This review entitled “Dietary polyphenols in metabolic diseases and neurodegeneration: Molecular targets on autophagy and biological effects” by Ana García-Aguilar et al. describes about the roles of dietary polyphenol in regulation of autophagy regarding pathological contest. The current situation of this topic is well and widely summarized and quite informative to the readers. Some minor points should be amended before publication.

line 52, plasmatic should be plasma

Line 184-192 Microautophagy and CMA are not mentioned hereafter, so should be removed. It is confusing to the readers.

Line 257, alfa cell?

Line 346 and so on. Amyloid alfa?

Line 434, What is CR?

Line446−452, Incomplete

Author Response

This review entitled “Dietary polyphenols in metabolic diseases and neurodegeneration: Molecular targets on autophagy and biological effects” by Ana García-Aguilar et al. describes about the roles of dietary polyphenol in regulation of autophagy regarding pathological contest. The current situation of this topic is well and widely summarized and quite informative to the readers. Some minor points should be amended before publication.

- Thanks a lot to this reviewer for her/his comments. We are profoundly grateful for them

line 52, plasmatic should be plasma

- We have done

Line 184-192 Microautophagy and CMA are not mentioned hereafter, so should be removed. It is confusing to the readers.

- We have done

Line 257, alfa cell?

- We have corrected

Line 346 and so on. Amyloid alfa?

- We have corrected

Line 434, What is CR?

- We have corrected

Line446−452, Incomplete

- We have corrected the funding section and we have removed acknowledgements section in the revised version of the manuscript.

Round 2

Reviewer 2 Report

The manuscript (Ms) by Garcia-Aguilar et al., has been improved considering the scientific content as the authors introduced most of the requested changes, but the Ms requires further major effort to meet the standards of Antioxidants. Several new sections of the review are chaotic and characterized by grammar or English style problems that make the Ms difficult to follow. The authors should deal with the following problems:

Major issues:

  1. The introduction has been improved, but it is still far from being acceptable. In many cases the authors have just added some new text (also redundant) without putting enough afford to organize and link better different paragraphs and develop on the main topics.
  2. They also left unnecessary paragraphs like “definition of apoptosis” (lines 44-52). Beside the fact that the first sentence is incorrect (necrosis is also a common cell damage caused by oxidative stress), it is not clear why it would be relevant to introduce the term “apoptosis”.
  3. The discussion of UPS (lines 99 to 103) is also redundant and distracts the reader from the main topic “autophagy”. In fact, UPS is only described in the introduction, while it is not mentioned in the other paragraphs as modulated by polyphenols.
  4. The term “autophagy” has been anticipated in the introduction of the revised Ms as suggested for the revision in p. 2, but the authors do not even mention that the enhanced autophagy is an important cellular mechanism underlying the beneficial effects of PPHs.
  5. The crosstalk between oxidative stress, autophagy and antioxidant effects of polyphenols, should be introduced briefly with 1 or 2 sentences in the introduction. Nevertheless, there is far more to say about this topic than what is written in lines 108 to 114, this part should be moved and discussed separately in the chapter 2. For discussion see: Janda et al. 2012 Mol. Neurobiol.; Costa LG et al. 2016, Oxid Med Cell Longev; Peña-Oyarzun D et al. 2018, Free Radic Biol Med.
  6. The section about polyphenol (PPH) classification and their role in plants and pharmacological activities is chaotic: two paragraphs in 2 different locations, which are partially repetitive (lines 58 to 87 and 115 to 136). These paragraphs should be organized in one paragraph. They also do not deliver a clear picture on how PPHs are classified (lines 58 to 76) and how they explicate their antioxidant function (line 85-87). Lines 58-76: Please improve English usage, organize better the ideas, and focus on what is relevant.
  7. Lines 85-87: Redox proprieties of polyphenols are heavily debated in the field, therefore please develop this point with a sufficient background and put relevant citations. See: 1) Forman, H.J.; et al. How do nutritional antioxidants really work (…). Free Radic Biol Med 2014, 66, 24-35, and 2) Franco, R. et al. Hormetic and Mitochondria-Related Mechanisms of Antioxidant Action of Phytochemicals. Antioxidants (Basel) 2019, 8.
  8. Correct, please, the title! It should be: Dietary polyphenols in metabolic and neurodegenerative diseases: Molecular targets in autophagy and biological effects.
  9. The section 2 (lines 142 to 186) regarding basic mechanisms of autophagosome formation should be reduced to a minimum, first because it is not relevant to modulation of autophagy by polyphenols and second, because there are hundreds of reviews that dedicate a specific attention to this topic. In addition, the authors describe the autophagosome formation again in the figure 2, while the PPH are known to modulate autophagy at upstream signaling steps, as correctly described in lines 242-272.
  10. The table 1: “The main biological effects of dietary polyphenols” is too generic. It should be limited only to the examples of autophagic activation by polyphenols in the context of health benefits in vivo (neurodegenerative, metabolic and cardiovascular diseases). The authors also should prepare another table reporting the major mechanisms or signaling pathways activated by polyphenols and involved in autophagy stimulation both in vitro or in vivo papers. Such a table would be very useful for the reader to find the “molecular targets in autophagy”. In addition the list of papers of the table 1 seems to be relatively poor considering that Pubmed reports 390 original articles with the terms “polyphenol” and “autophagy”.
  11. The Figure 3 should be consistent with the molecular targets of autophagy reported in lines 242 to 272.This paragraph should also mention: Pietrocola F, PlosOne 2012, “Pro-autophagic polyphenols reduce the acetylation of cytoplasmic proteins”.

Minor issues:

  1. Line 43: The reference 3 is not appropriate, Please, cite at least 3 more relevant papers that demonstrate the activation of different signaling pathways by ROS.
  2. The term “To sum up” is not appropriate since the authors do not mention anything about the purpose of this review.
  3. The authors should also mention also other papers related to the effects of polyphenols on NAFLD, such as: Ding S et al., 2017, PlosOne. Resveratrol and caloric restriction prevent hepatic steatosis by regulating SIRT1-autophagy; Lascala A et al., 2018, J Nutr Biochem. (also relevant to 94); Zhang Y et al., 2015, Mol Nutr Food Res. Resveratrol improves hepatic steatosis by inducing autophagy through the cAMP
  • 4. The added new (red) parts of the review are full of grammar mistakes and bad English usage. They were also present in the first version of Ms, but it was not useless to correct them before the major revision. Just some examples:
  1. Obesity, one of the main factors to develop T2DM, has been involved with an alteration in autophagy. (should be: “…. T2DM, has been implicated in ….”
  2. 403-403 “….. associated with an increased in the autophagic flux.”

Author Response

The manuscript (Ms) by Garcia-Aguilar et al., has been improved considering the scientific content as the authors introduced most of the requested changes, but the Ms requires further major effort to meet the standards of Antioxidants. Several new sections of the review are chaotic and characterized by grammar or English style problems that make the Ms difficult to follow. The authors should deal with the following problems:

Major issues:

  1. The introduction has been improved, but it is still far from being acceptable. In many cases the authors have just added some new text (also redundant) without putting enough afford to organize and link better different paragraphs and develop on the main topics.

- The authors thank this comment and the text has been reviewed accordingly.

     2. They also left unnecessary paragraphs like “definition of apoptosis” (lines 44-52). Beside the fact that the first sentence is incorrect (necrosis is also a common cell damage caused by oxidative stress), it is not clear why it would be relevant to introduce the term “apoptosis”.

- The authors thank this comment and the text has been reviewed accordingly.

      3. The discussion of UPS (lines 99 to 103) is also redundant and distracts the reader from the main topic “autophagy”. In fact, UPS is only described in the introduction, while it is not mentioned in the other paragraphs as modulated by polyphenols.

- The authors are totally agreed with this comment and the paragraph corresponding to the UPS definition has been deleted. We also mention in the revised manuscript that in this review we will focus on the autophagic process (line 126)

      4. The term “autophagy” has been anticipated in the introduction of the revised Ms as suggested for the revision in p. 2, but the authors do not even mention that the enhanced autophagy is an important cellular mechanism underlying the beneficial effects of PPHs.

- The authors are agreed with this comment, and we have highlighted the importance of autophagy activation by PPHs as one of the main mechanisms of their beneficial effects (lines 102-103)

       5. The crosstalk between oxidative stress, autophagy and antioxidant effects of polyphenols, should be introduced briefly with 1 or 2 sentences in the introduction.

- The authors are totally agreed with this comment and we have included a sentence to introduce the topic (lines 56-58)

       6. Nevertheless, there is far more to say about this topic than what is written in lines 108 to 114, this part should be moved and discussed separately in the chapter 2. For discussion see: Janda et al. 2012 Mol. Neurobiol.; Costa LG et al. 2016, Oxid Med Cell Longev; Peña-Oyarzun D et al. 2018, Free Radic Biol Med.

- The authors thank this comment and the text has been reviewed accordingly (see 222-228) and the references suggested by the reviewer have been incorporated in the revised version of the manuscript.

       7. The section about polyphenol (PPH) classification and their role in plants and pharmacological activities is chaotic: two paragraphs in 2 different locations, which are partially repetitive (lines 58 to 87 and 115 to 136). These paragraphs should be organized in one paragraph. They also do not deliver a clear picture on how PPHs are classified (lines 58 to 76) and how they explicate their antioxidant function (line 85-87). Lines 58-76: Please improve English usage, organize better the ideas, and focus on what is relevant.

- The authors thank this comment and the text has been reviewed accordingly.

       8. Lines 85-87: Redox proprieties of polyphenols are heavily debated in the field, therefore please develop this point with a sufficient background and put relevant citations. See: 1) Forman, H.J.; et al. How do nutritional antioxidants really work (…). Free Radic Biol Med 201466, 24-35, and 2) Franco, R. et al. Hormetic and Mitochondria-Related Mechanisms of Antioxidant Action of Phytochemicals. Antioxidants (Basel) 20198.

- The authors are partially agreed with this comment. As the reviewer pointed out, the redox properties of polyphenols are highly discussed and this point is not the main justification of this manuscript, which the principal aim is to describe the molecular targets of PPHs in autophagy. In any case, the authors have included and briefly discussed the references by Forman et al., 2014 and Franco et al., 2019 in the revised version of the manuscript.

       9. Correct, please, the title! It should be: Dietary polyphenols in metabolic and neurodegenerative diseases: Molecular targets in autophagy and biological effects.

- Thanks a lot for this comment. As suggested by the reviewer, the title has been corrected in the revised version of the manuscript

      10. The section 2 (lines 142 to 186) regarding basic mechanisms of autophagosome formation should be reduced to a minimum, first because it is not relevant to modulation of autophagy by polyphenols and second, because there are hundreds of reviews that dedicate a specific attention to this topic. In addition, the authors describe the autophagosome formation again in the figure 2, while the PPH are known to modulate autophagy at upstream signaling steps, as correctly described in lines 242-272.

- The authors thank this comment to the reviewer and, accordingly, the text has been modified.

       11. The table 1: “The main biological effects of dietary polyphenols” is too generic. It should be limited only to the examples of autophagic activation by polyphenols in the context of health benefits in vivo (neurodegenerative, metabolic and cardiovascular diseases). The authors also should prepare another table reporting the major mechanisms or signaling pathways activated by polyphenols and involved in autophagy stimulation both in vitro or in vivo papers. Such a table would be very useful for the reader to find the “molecular targets in autophagy”. In addition the list of papers of the table 1 seems to be relatively poor considering that Pubmed reports 390 original articles with the terms “polyphenol” and “autophagy”.

- We thank the reviewer for this comment. We have changed the title of the table 1, as suggested by this reviewer. Regarding the second point of the reviewer, instead of adding a new table to the revised version of the manuscript, we have decided to include in the new figure 3, the major signaling pathways induced by polyphenols, involving autophagy. In relation to the last point highlighted by the reviewer, it is true that there are many papers (382 papers) published in the last 5 years when you make the search for polyphenol and autophagy. However, the scope of the present review is the effect of polyphenols in autophagy in relationship with metabolic disease (then, the number is reduced to 40 papers) or cardiovascular disease (in this case there are 51 papers) and neurodegenerative diseases (in this case there are 49 papers). We would like to highlight that, in many cases, the molecular mechanisms of action of polyphenols are shared in the different pathologies and, the authors think that in the new figure 3, are included the main effectors regarding this topic.

      12. The Figure 3 should be consistent with the molecular targets of autophagy reported in lines 242 to 272.

- Thanks to the reviewer for his/her comment. We have modified the figure 3 in order to be consistent with the Ms. For this reason, we have included in this new figure 3 the main molecular targets of autophagy regulation reported throughout the manuscript including CaMKKβ, AKT, LKB1, BECN1, TFEB, FOXO3a and miRNA-18a-5p. The figure legend of new figure 3 has been also modified in the manuscript as well.

       13. This paragraph should also mention: Pietrocola F, PlosOne 2012, “Pro-autophagic polyphenols reduce the acetylation of cytoplasmic proteins”.

- As suggested by the reviewer, we have incorporated a comment of this paper and included in the bibliography in the revised version of the manuscript.

Minor issues:

  1. Line 43: The reference 3 is not appropriate, Please, cite at least 3 more relevant papers that demonstrate the activation of different signaling pathways by ROS.

- Thanks a lot to the reviewer for this comment. We have incorporated new references regarding the activation of ROS-mediated signaling pathways.

    2. The term “To sum up” is not appropriate since the authors do not mention anything about the purpose of this review.

- Thanks a lot to the reviewer for this comment. We have corrected in the revised manuscript.

    3. The authors should also mention also other papers related to the effects of polyphenols on NAFLD, such as: Ding S et al., 2017, PlosOne. Resveratrol and caloric restriction prevent hepatic steatosis by regulating SIRT1-autophagy; Lascala A et al., 2018, J Nutr Biochem. (also relevant to 94); Zhang Y et al., 2015, Mol Nutr Food Res. Resveratrol improves hepatic steatosis by inducing autophagy through the cAMP

- Thanks a lot to this reviewer for the suggestion. We have incorporated and discussed these references in the revised version of the manuscript.

    4. The added new (red) parts of the review are full of grammar mistakes and bad English usage. They were also present in the first version of Ms, but it was not useless to correct them before the major revision. Just some examples:

  1. Obesity, one of the main factors to develop T2DM, has been involved with an alteration in autophagy. (should be: “…. T2DM, has been implicated in ….”
  2. 403-403 “….. associated with an increased in the autophagic flux.”

- The authors thank this comment and the text has been reviewed accordingly.

Round 3

Reviewer 2 Report

The manuscript (Ms) by Garcia-Aguilar et al., antioxidants-1002268-v3 has been improved only in part since the authors have introduced only a portion of the requested changes. In addition, some changes are sloppy and superficial, containing grammar or style mistakes. As a consequence, the Ms continues to be difficult to follow. Finally, some sentences reveal a poor understanding of certain areas of research that authors tried to discuss.

I really hope that this time the authors address all the issues and comments seriously. The reviewer will not be able to accept the review if these major problems persist. The authors should deal with the problems indicated in the full review report attached below

Major issues:
1. The introduction has been improved, but it is still far from being acceptable. In many cases the authors have just added some new text (also redundant) without putting enough afford to organize and link better different paragraphs and develop on the main topics.
- The authors thank this comment, and the text has been reviewed accordingly.
+ C1A: Many efforts to link paragraphs have failed, as authors used unconvincing sentences like in line 46:
In addition, autophagy has been considered as a non-apoptotic mechanism of eukaryotic cells…” This sentence is simply wrong, as autophagy is not “a non-apoptotic mechanism ….” See the reply to p. 3
Line 38: “The main response ….” should be a new paragraph. Please see the comment C2B

2. They also left unnecessary paragraphs like “definition of apoptosis” (lines 44-52). Beside the fact that the first sentence is incorrect (necrosis is also a common cell damage caused by oxidative stress), it is not clear why it would be relevant to introduce the term “apoptosis”.
- The authors thank this comment and the text has been reviewed accordingly.
+ C2A: The text is shorter, but the problem remains unsolved!! In lines 37-40, the authors describe what is apoptosis as if it were the only cell death type induced by ROS, while it is NOT, and the authors should mention other types of cell death that might be triggered by excessive ROS. Apoptosis and necrosis are viewed as opposite extremes on a spectrum of cell death and can occur simultaneously in the same tissue in response to oxidative stress (Morris et al. 2017). Moran Benhar in Antioxidants 2020 writes: “Growing evidence indicates that redox modifications of cysteine residues in proteins are involved in the regulation of multiple cell death modalities, including apoptosis, necroptosis and pyroptosis.”
+ C2B: Line 37-39: “The main response to tissue damage caused by intense oxidative stress is uncontrolled apoptosis as a highly regulated programmed cell death pathway”. This sentence contains an intrinsic contradiction: “uncontrolled apoptosis ……”. It cannot be uncontrolled if it is highly programmed ……..??
+ C2C: The authors continue describing “apoptosis” in lines 40-45. It is not necessary for the purpose of this review. Instead, they should state that oxy stress triggers different types of cell death and then briefly characterize the differences.

3. The discussion of UPS (lines 99 to 103) is also redundant and distracts the reader from the main topic “autophagy”. In fact, UPS is only described in the introduction, while it is not mentioned in the other paragraphs as modulated by polyphenols.
- The authors are totally agreed with this comment and the paragraph corresponding to the UPS definition has been deleted. We also mention in the revised manuscript that in this review we will focus on the autophagic process (line 126).
+ C3A: The authors should not eliminate the first sentence in line 96 – 98 of Ms v2. “Moreover, in response to cellular stress, eukaryotic cells can activate several physiological degradative pathways … autophagy.” and move it to line 46 instead of the phrase “In addition, autophagy has been considered as a non-apoptotic mechanism of eukaryotic cells…” which is simply wrong, as autophagy is not “a non-apoptotic mechanism.”

4. The term “autophagy” has been anticipated in the introduction of the revised Ms as suggested for the revision in p. 2, but the authors do not even mention that the enhanced autophagy is an important cellular mechanism underlying the beneficial effects of PPHs.
- The authors are agreed with this comment, and we have highlighted the importance of autophagy activation by PPHs as one of the main mechanisms of their beneficial effects (lines 102-103)
+ C4: It is OK, but please reformulate or split the sentence 97-103, as it is 7 lines long. Line 98: “… to measure but include actions such as hormesis, defined …..“ the term “Actions such as” is redundant.

5. The crosstalk between oxidative stress, autophagy and antioxidant effects of polyphenols, should be introduced briefly with 1 or 2 sentences in the introduction.
- The authors are totally agreed with this comment and we have included a sentence to introduce the topic (lines 56-58)
+ C5A: Well, the indicated sentence does not fulfill the point, as another sentence is needed that explains the cross-talk i.e. Although oxidative stress (OS) stimulates autophagy in different cellular systems, it has been shown that a prolonged OS inhibits autophagy …. (Limanaqi F, Biagioni F,… Antioxidants (Basel). 2020; Janda E, Lascala A,….. Autophagy. 2015).
+ C5B: In addition, please correct the grammar of the sentence “Recently, there is increasing evidence in the potential health benefits of dietary plant…”.

6. Nevertheless, there is far more to say about this topic than what is written in lines 108 to 114, this part should be moved and discussed separately in the chapter 2. For discussion see: Janda et al. 2012 Mol. Neurobiol.; Costa LG et al. 2016, Oxid Med Cell Longev; Peña-Oyarzun D et al. 2018, Free Radic Biol Med.
- The authors thank this comment and the text has been reviewed accordingly (see 222-228) and the references suggested by the reviewer have been incorporated in the revised version of the manuscript.
+ C6: Well, it is just one mechanism of the OS-autophagy cross-talk. The authors should mention the existence of other mechanisms with one or two sentences.

7. The section about polyphenol (PPH) classification and their role in plants and pharmacological activities is chaotic: two paragraphs in 2 different locations, which are partially repetitive (lines 58 to 87 and 115 to 136). These paragraphs should be organized in one paragraph. They also do not deliver a clear picture on how PPHs are classified (lines 58 to 76) and how they explicate their antioxidant function (line 85-87). Lines 58-76: Please improve English usage, organize better the ideas, and focus on what is relevant.
- The authors thank this comment and the text has been reviewed accordingly.
+ C7: This is fine, except sentence in lines 97-103. See the comment to p.4.

8. Lines 85-87: Redox proprieties of polyphenols are heavily debated in the field, therefore please develop this point with a sufficient background and put relevant citations. See: 1) Forman, H.J.; et al. How do nutritional antioxidants really work (…). Free Radic Biol Med 2014, 66, 24-35, and 2) Franco, R. et al. Hormetic and Mitochondria-Related Mechanisms of Antioxidant Action of Phytochemicals. Antioxidants (Basel) 2019, 8.
- The authors are partially agreed with this comment. As the reviewer pointed out, the redox properties of polyphenols are highly discussed and this point is not the main justification of this manuscript, which the principal aim is to describe the molecular targets of PPHs in autophagy. In any case, the authors have included and briefly discussed the references by Forman et al., 2014 and Franco et al., 2019 in the revised version of the manuscript.
+ This is OK.

9. Correct, please, the title! It should be: Dietary polyphenols in metabolic and neurodegenerative diseases: Molecular targets in autophagy and biological effects.
- Thanks a lot for this comment. As suggested by the reviewer, the title has been corrected in the revised version of the manuscript
+ This is OK.

10. The section 2 (lines 142 to 186) regarding basic mechanisms of autophagosome formation should be reduced to a minimum, first because it is not relevant to modulation of autophagy by polyphenols and second, because there are hundreds of reviews that dedicate a specific attention to this topic. In addition, the authors describe the autophagosome formation again in the figure 2, while the PPH are known to modulate autophagy at upstream signaling steps, as correctly described in lines 242-272.
- The authors thank this comment to the reviewer and, accordingly, the text has been modified.
+ C10A: Further reduction can be achieved. The first sentence in line 125-126 should be deleted. The description of autophagosome formation in lines 145-155 should be further reduced “The present review focuses on the macroautophagy process (hereafter referred to as autophagy) which takes place in the following main four steps: (1) Initiation, (2) the nucleation step … as described in detail in Fig. 2”. The description of autophagy steps in the legend of Fig. 2 should indicated by numbers and detailed as in the v3 text.
+ C10B: Please check the grammar: Line 132 should be “stressors” or “stresses”. Line 137: should be “involve”. Line 138 “considered” is not correct, please rephrase.

11. The table 1: “The main biological effects of dietary polyphenols” is too generic. It should be limited only to the examples of autophagic activation by polyphenols in the context of health benefits in vivo (neurodegenerative, metabolic and cardiovascular diseases). The authors also should prepare another table reporting the major mechanisms or signaling pathways activated by polyphenols and involved in autophagy stimulation both in vitro or in vivo papers. Such a table would be very useful for the reader to find the “molecular targets in autophagy”. In addition the list of papers of the table 1 seems to be relatively poor considering that Pubmed reports 390 original articles with the terms “polyphenol” and “autophagy”.
- We thank the reviewer for this comment. We have changed the title of the table 1, as suggested by this reviewer. Regarding the second point of the reviewer, instead of adding a new table to the revised version of the manuscript, we have decided to include in the new figure 3, the major signaling pathways induced by polyphenols, involving autophagy. In relation to the last point highlighted by the reviewer, it is true that there are many papers (382 papers) published in the last 5 years when you make the search for polyphenol and autophagy. However, the scope of the present review is the effect of polyphenols in autophagy in relationship with metabolic disease (then, the number is reduced to 40 papers) or cardiovascular disease (in this case there are 51 papers) and neurodegenerative diseases (in this case there are 49 papers). We would like to highlight that, in many cases, the molecular mechanisms of action of polyphenols are shared in the different pathologies and, the authors think that in the new figure 3, are included the main effectors regarding this topic.
+ C11A: This reviewer agrees with respect to the Fig. 3 modification. However, the table 3 should be expanded. The authors mentioned that flavonoids are the largest polyphenol group, yet they are poorly represented in the table 3. Note that several important flavonoids are missing like apigenin, luteolin, baicalein and others, while quercetin, the most studied flavonoid, is cited only once out of 173 citations in PubMed.
+ C11B: In addition, there are more than 1000 papers when two terms “autophagy” and “flavonoid” are searched for, and majority of them are about modulation of autophagy by flavonoids. Please add at least apigenin, luteolin, baicalein to the table with at least 5 citations to each flavonoid and representatively higher number to quercetin. Other examples of stilbenes, catechins tannins and especially anthocyanidins (55 citations in Pubmed together with autophagy) can be added as well.

12. The Figure 3 should be consistent with the molecular targets of autophagy reported in lines 242 to 272.
- Thanks to the reviewer for his/her comment. We have modified the figure 3 in order to be consistent with the Ms. For this reason, we have included in this new figure 3 the main molecular targets of autophagy regulation reported throughout the manuscript including CaMKKβ, AKT, LKB1, BECN1, TFEB, FOXO3a and miRNA-18a-5p. The figure legend of new figure 3 has been also modified in the manuscript as well.
+ OK

13. This paragraph should also mention: Pietrocola F, PlosOne 2012, “Pro-autophagic polyphenols reduce the acetylation of cytoplasmic proteins”.
- As suggested by the reviewer, we have incorporated a comment of this paper and included in the bibliography in the revised version of the manuscript.
+ C13A: OK, but the authors should mention the enzyme involved.

Minor issues:
1. Line 43: The reference 3 is not appropriate, Please, cite at least 3 more relevant papers that demonstrate the activation of different signaling pathways by ROS.
- Thanks a lot to the reviewer for this comment. We have incorporated new references regarding the activation of ROS-mediated signaling pathways.
+ OK
2. The term “To sum up” is not appropriate since the authors do not mention anything about the purpose of this review.
- Thanks a lot to the reviewer for this comment. We have corrected in the revised manuscript.
+ OK
3. The authors should mention also other papers related to the effects of polyphenols on NAFLD, such as: Ding S et al., 2017, PlosOne. Resveratrol and caloric restriction prevent hepatic steatosis by regulating SIRT1-autophagy; Lascala A et al., 2018, J Nutr Biochem. (also relevant to 94); Zhang Y et al., 2015, Mol Nutr Food Res. Resveratrol improves hepatic steatosis by inducing autophagy through the cAMP
- Thanks a lot to this reviewer for the suggestion. We have incorporated and discussed these references in the revised version of the manuscript.
+ OK
4. The added new (red) parts of the review are full of grammar mistakes and bad English usage. They were also present in the first version of Ms, but it was not useless to correct them before the major revision. Just some examples:
1. Obesity, one of the main factors to develop T2DM, has been involved with an alteration in autophagy. (should be: “…. T2DM, has been implicated in ….”
2. 403-403 “….. associated with an increased in the autophagic flux.”
- The authors thank this comment and the text has been reviewed accordingly.
+ Further minor points to the report 3
5. Correct “stylbene/s” to a more commonly accepted term “stilbene/s” throughout the manuscript.
6. Lines 27-28: Please correct as indicated: “One of the consequences of the normal function in living organisms is the intracellular production of free radicals reactive oxygen species (ROS) in
significant amounts, mainly located (found) in the cytosol, mitochondria, lysosomes, peroxisomes and epithelial membranes. “
7. The introduction of the new term ROS (or RNS) is necessary in the introduction when oxidative stress issue is introduced (see above as it is suggested in. p.6.) and not in line 197.
8. Please correct English and style errors throughout as they are numerous and difficult to list.

Author Response

Major issues:

  1. The introduction has been improved, but it is still far from being acceptable. In many cases the authors have just added some new text (also redundant) without putting enough afford to organize and link better different paragraphs and develop on the main topics.

- The authors thank this comment, and the text has been reviewed accordingly.

+ C1A: Many efforts to link paragraphs have failed, as authors used unconvincing sentences like in line 46:In addition, autophagy has been considered as a non-apoptotic mechanism of eukaryotic cells…” This sentence is simply wrong, as autophagy is not “a non-apoptotic mechanism ….” See the reply to p. 3

- The authors have corrected the sentences indicated by this reviewer in order to improve the understanding. The changes are highlighted in red color in the manuscript (page 2, lines 46 and 47)

Line 38: “The main response ….” should be a new paragraph. Please see the comment C2B

The sentence has been corrected and starts a new paragraph.

  1. They also left unnecessary paragraphs like “definition of apoptosis” (lines 44-52). Beside the fact that the first sentence is incorrect (necrosis is also a common cell damage caused by oxidative stress), it is not clear why it would be relevant to introduce the term “apoptosis”.

- The authors thank this comment and the text has been reviewed accordingly.

+ C2A: The text is shorter, but the problem remains unsolved!! In lines 37-40, the authors describe what is apoptosis as if it were the only cell death type induced by ROS, while it is NOT, and the authors should mention other types of cell death that might be triggered by excessive ROS. Apoptosis and necrosis are viewed as opposite extremes on a spectrum of cell death and can occur simultaneously in the same tissue in response to oxidative stress (Morris et al. 2017). Moran Benhar in Antioxidants 2020 writes: “Growing evidence indicates that redox modifications of cysteine residues in proteins are involved in the regulation of multiple cell death modalities, including apoptosis, necroptosis and pyroptosis.”

- The authors completely agree with the reviewer in the fact that there are several types of cell death by excessive ROS. However, we have decided not to include any concept for which no reference will be included later in the manuscript, in order not to distract the reader from the main topic of the review.

+ C2B: Line 37-39: “The main response to tissue damage caused by intense oxidative stress is uncontrolled apoptosis as a highly regulated programmed cell death pathway”. This sentence contains an intrinsic contradiction: “uncontrolled apoptosis ……”. It cannot be uncontrolled if it is highly programmed ……..??

- The authors have corrected the sentence for eliminating the contradiction.

+ C2C: The authors continue describing “apoptosis” in lines 40-45. It is not necessary for the purpose of this review. Instead, they should state that oxy stress triggers different types of cell death and then briefly characterize the differences.

- The authors think that a brief mention of apoptosis, which is one of the main mechanisms for cell death, is still necessary. Furthermore, the main topic of this review is to analyze the role of different dietary polyphenols in several diseases (metabolic and neurodegenerative diseases) with special emphasis in the molecular targets in autophagy. In this regard, apoptosis and autophagy are interconnected as it has been described and studied in many papers (see new references 16 and 17 in the third revised version of the manuscript).

  1. The discussion of UPS (lines 99 to 103) is also redundant and distracts the reader from the main topic “autophagy”. In fact, UPS is only described in the introduction, while it is not mentioned in the other paragraphs as modulated by polyphenols.

- The authors are totally agreed with this comment and the paragraph corresponding to the UPS definition has been deleted. We also mention in the revised manuscript that in this review we will focus on the autophagic process (line 126).

+ C3A: The authors should not eliminate the first sentence in line 96 – 98 of Ms v2. “Moreover, in response to cellular stress, eukaryotic cells can activate several physiological degradative pathways … autophagy.” and move it to line 46 instead of the phrase “In addition, autophagy has been considered as a non-apoptotic mechanism of eukaryotic cells…” which is simply wrong, as autophagy is not “a non-apoptotic mechanism.”

- The first sentence of the previous version of the manuscript has been included again. In addition, we have moved the sentence to line 46 of page 2, as suggested by the reviewer.

  1. The term “autophagy” has been anticipated in the introduction of the revised Ms as suggested for the revision in p. 2, but the authors do not even mention that the enhanced autophagy is an important cellular mechanism underlying the beneficial effects of PPHs.

- The authors are agreed with this comment, and we have highlighted the importance of autophagy activation by PPHs as one of the main mechanisms of their beneficial effects (lines 102-103)

+ C4: It is OK, but please reformulate or split the sentence 97-103, as it is 7 lines long. Line 98: “… to measure but include actions such as hormesis, defined …..“ the term “Actions such as” is redundant.

- We have changed the sentence in line 102 according to the reviewer´s comment.

  1. The crosstalk between oxidative stress, autophagy and antioxidant effects of polyphenols, should be introduced briefly with 1 or 2 sentences in the introduction.

- The authors are totally agreed with this comment and we have included a sentence to introduce the topic (lines 56-58)

+ C5A: Well, the indicated sentence does not fulfill the point, as another sentence is needed that explains the cross-talk i.e. Although oxidative stress (OS) stimulates autophagy in different cellular systems, it has been shown that a prolonged OS inhibits autophagy …. (Limanaqi F, Biagioni F,… Antioxidants (Basel). 2020; Janda E, Lascala A,….. Autophagy. 2015).

- We have added a new sentence in line 50-51 in page 2 as suggested by the reviewer.

+C5B: In addition, please correct the grammar of the sentence “Recently, there is increasing evidence in the potential health benefits of dietary plant…”.

- We have changed the sentence in line 60-62 in page 2 as suggested by the reviewer.

  1. Nevertheless, there is far more to say about this topic than what is written in lines 108 to 114, this part should be moved and discussed separately in the chapter 2. For discussion see: Janda et al. 2012 Mol. Neurobiol.; Costa LG et al. 2016, Oxid Med Cell Longev; Peña-Oyarzun D et al. 2018, Free Radic Biol Med.

- The authors thank this comment and the text has been reviewed accordingly (see 222-228) and the references suggested by the reviewer have been incorporated in the revised version of the manuscript.

+ C6: Well, it is just one mechanism of the OS-autophagy cross-talk. The authors should mention the existence of other mechanisms with one or two sentences.

- Following the reviewer’s suggestions, we have mentioned that there are other mechanisms involved in autophagy regulation by OS in lines 219-220 of the revised ms.

  1. The section about polyphenol (PPH) classification and their role in plants and pharmacological activities is chaotic: two paragraphs in 2 different locations, which are partially repetitive (lines 58 to 87 and 115 to 136). These paragraphs should be organized in one paragraph. They also do not deliver a clear picture on how PPHs are classified (lines 58 to 76) and how they explicate their antioxidant function (line 85-87). Lines 58-76: Please improve English usage, organize better the ideas, and focus on what is relevant.

- The authors thank this comment and the text has been reviewed accordingly.

+ C7: This is fine, except sentence in lines 97-103. See the comment to p.4.

- We have changed the sentence in line 102 according to the reviewer´s comment.

  1. Lines 85-87: Redox proprieties of polyphenols are heavily debated in the field, therefore please develop this point with a sufficient background and put relevant citations. See: 1) Forman, H.J.; et al. How do nutritional antioxidants really work (…). Free Radic Biol Med 2014, 66, 24-35, and 2) Franco, R. et al. Hormetic and Mitochondria-Related Mechanisms of Antioxidant Action of Phytochemicals. Antioxidants (Basel) 2019, 8.

- The authors are partially agreed with this comment. As the reviewer pointed out, the redox properties of polyphenols are highly discussed and this point is not the main justification of this manuscript, which the principal aim is to describe the molecular targets of PPHs in autophagy. In any case, the authors have included and briefly discussed the references by Forman et al., 2014 and Franco et al., 2019 in the revised version of the manuscript.

+ This is OK.

  1. Correct, please, the title! It should be: Dietary polyphenols in metabolic and neurodegenerative diseases: Molecular targets in autophagy and biological effects.

- Thanks a lot for this comment. As suggested by the reviewer, the title has been corrected in the revised version of the manuscript

+ This is OK.

  1. The section 2 (lines 142 to 186) regarding basic mechanisms of autophagosome formation should be reduced to a minimum, first because it is not relevant to modulation of autophagy by polyphenols and second, because there are hundreds of reviews that dedicate a specific attention to this topic. In addition, the authors describe the autophagosome formation again in the figure 2, while the PPH are known to modulate autophagy at upstream signaling steps, as correctly described in lines 242-272.

- The authors thank this comment to the reviewer and, accordingly, the text has been modified.

+ C10A: Further reduction can be achieved. The first sentence in line 125-126 should be deleted. The description of autophagosome formation in lines 145-155 should be further reduced “The present review focuses on the macroautophagy process (hereafter referred to as autophagy) which takes place in the following main four steps: (1) Initiation, (2) the nucleation step … as described in detail in Fig. 2”. The description of autophagy steps in the legend of Fig. 2 should indicated by numbers and detailed as in the v3 text.

- We have changed the text accordingly.

+ C10B: Please check the grammar: Line 132 should be “stressors” or “stresses”. Line 137: should be “involve”. Line 138 “considered” is not correct, please rephrase.

- We have changed the text accordingly.

  1. The table 1: “The main biological effects of dietary polyphenols” is too generic. It should be limited only to the examples of autophagic activation by polyphenols in the context of health benefits in vivo (neurodegenerative, metabolic and cardiovascular diseases). The authors also should prepare another table reporting the major mechanisms or signaling pathways activated by polyphenols and involved in autophagy stimulation both in vitro or in vivo papers. Such a table would be very useful for the reader to find the “molecular targets in autophagy”. In addition the list of papers of the table 1 seems to be relatively poor considering that Pubmed reports 390 original articles with the terms “polyphenol” and “autophagy”.

- We thank the reviewer for this comment. We have changed the title of the table 1, as suggested by this reviewer. Regarding the second point of the reviewer, instead of adding a new table to the revised version of the manuscript, we have decided to include in the new figure 3, the major signaling pathways induced by polyphenols, involving autophagy. In relation to the last point highlighted by the reviewer, it is true that there are many papers (382 papers) published in the last 5 years when you make the search for polyphenol and autophagy. However, the scope of the present review is the effect of polyphenols in autophagy in relationship with metabolic disease (then, the number is reduced to 40 papers) or cardiovascular disease (in this case there are 51 papers) and neurodegenerative diseases (in this case there are 49 papers). We would like to highlight that, in many cases, the molecular mechanisms of action of polyphenols are shared in the different pathologies and, the authors think that in the new figure 3, are included the main effectors regarding this topic.

+ C11A: This reviewer agrees with respect to the Fig. 3 modification. However, the table 3 should be expanded. The authors mentioned that flavonoids are the largest polyphenol group, yet they are poorly represented in the table 3. Note that several important flavonoids are missing like apigenin, luteolin, baicalein and others, while quercetin, the most studied flavonoid, is cited only once out of 173 citations in PubMed.

- We have changed the Table content according the reviewer’s suggestions.

+ C11B: In addition, there are more than 1000 papers when two terms “autophagy” and “flavonoid” are searched for, and majority of them are about modulation of autophagy by flavonoids. Please add at least apigenin, luteolin, baicalein to the table with at least 5 citations to each flavonoid and representatively higher number to quercetin. Other examples of stilbenes, catechins tannins and especially anthocyanidins (55 citations in Pubmed together with autophagy) can be added as well.

- We have modified the table content accordingly.

  1. The Figure 3 should be consistent with the molecular targets of autophagy reported in lines 242 to 272.

- Thanks to the reviewer for his/her comment. We have modified the figure 3 in order to be consistent with the Ms. For this reason, we have included in this new figure 3 the main molecular targets of autophagy regulation reported throughout the manuscript including CaMKKβ, AKT, LKB1, BECN1, TFEB, FOXO3a and miRNA-18a-5p. The figure legend of new figure 3 has been also modified in the manuscript as well.

+ OK

  1. This paragraph should also mention: Pietrocola F, PlosOne 2012, “Pro-autophagic polyphenols reduce the acetylation of cytoplasmic proteins”.

- As suggested by the reviewer, we have incorporated a comment of this paper and included in the bibliography in the revised version of the manuscript.

+ C13A: OK, but the authors should mention the enzyme involved.

- The authors have mentioned the enzyme involved, according the reviewer’s suggestions.

Minor issues:

  1. Line 43: The reference 3 is not appropriate, Please, cite at least 3 more relevant papers that demonstrate the activation of different signaling pathways by ROS.

- Thanks a lot to the reviewer for this comment. We have incorporated new references regarding the activation of ROS-mediated signaling pathways.

+ OK

  1. The term “To sum up” is not appropriate since the authors do not mention anything about the purpose of this review.

- Thanks a lot to the reviewer for this comment. We have corrected in the revised manuscript.

+ OK

  1. The authors should mention also other papers related to the effects of polyphenols on NAFLD, such as: Ding S et al., 2017, PlosOne. Resveratrol and caloric restriction prevent hepatic steatosis by regulating SIRT1-autophagy; Lascala A et al., 2018, J Nutr Biochem. (also relevant to 94); Zhang Y et al., 2015, Mol Nutr Food Res. Resveratrol improves hepatic steatosis by inducing autophagy through the cAMP

- Thanks a lot to this reviewer for the suggestion. We have incorporated and discussed these references in the revised version of the manuscript.

+ OK

  1. The added new (red) parts of the review are full of grammar mistakes and bad English usage. They were also present in the first version of Ms, but it was not useless to correct them before the major revision. Just some examples:

           1. Obesity, one of the main factors to develop T2DM, has been involved with an alteration in autophagy. (should be: “…. T2DM, has been implicated in ….”

           2. 403-403 “….. associated with an increased in the autophagic flux.”

- The authors thank this comment and the text has been reviewed accordingly.

+ Further minor points to the report 3

  1. Correct “stylbene/s” to a more commonly accepted term “stilbene/s” throughout the manuscript.

- The term stylbenes has been corrected to stilbenes throughout the manuscript.

  1. Lines 27-28: Please correct as indicated: “One of the consequences of the normal function in living organisms is the intracellular production of free radicals reactive oxygen species (ROS) in significant amounts, mainly located (found) in the cytosol, mitochondria, lysosomes, peroxisomes and epithelial membranes. “

- The text in the manuscript has been corrected as indicated by the reviewer.

  1. The introduction of the new term ROS (or RNS) is necessary in the introduction when oxidative stress issue is introduced (see above as it is suggested in. p.6.) and not in line 197.

- The term ROS has been introduced in the Introduction section according the reviewer’s suggestion.

  1. Please correct English and style errors throughout as they are numerous and difficult to list.

- English language has been reviewed and corrected.

Round 4

Reviewer 2 Report

The authors have addressed carefully most of the issues pointed out by the referees. However, there are still numerous minor problems that need another round of revision:  

  1. The abstract is too generic and does not provide enough information about the content of the review. The only sentence that aims to make a synthesis of the content is confusing and not informative: “We report the main biological effects in relationship to autophagy regulation in response to different dietary polyphenols and its impact on metabolic and neurodegenerative diseases.” The authors should rewrite this sentence and add other 2 up to 3 sentences providing more information about the specific topics discussed in this work. The success of the review depends on the title and the abstract and a badly written summary will not attract the attention of potential readers.
  2. Lines 39 and 40. This sentence is still incorrect. If the authors do not want to mention other types of cell death, at least they should rephrase to “ The main response to tissue damage caused intense oxidative stress is cell death. The most studied type of cell death is apoptosis. It is runs through several steps, starting …”
  3. The authors should shorten the description of apoptosis in the introduction. 6 lines is too much for a topic barely mentioned in the main text.
  4. The section title: 4. Effect of polyphenols in cardiovascular complications” should be “… on cardiovascular diseases”, The term “complication” in medicine is restricted to “unwanted consequences of a primary disease”.
  5. Line 54-55: The sentence “Furthermore, there are positive as well as negative regulatory loops in different proteins involved in either apoptosis or autophagy (16,17).” is unclear and out of context with respect to the previous sentence” It should be rephrased.
  6. The manuscript is still rich in style and grammar errors. Just one example:

Line 60 and 221: “ .. increasing evidences demonstrate… ” should be “increasing evidence demonstrates”

The authors should carefully revise and eliminate all style and grammar errors, if they want to avoid further rounds of the revision.  

Author Response

The authors have addressed carefully most of the issues pointed out by the referees. However, there are still numerous minor problems that need another round of revision:  

The abstract is too generic and does not provide enough information about the content of the review. The only sentence that aims to make a synthesis of the content is confusing and not informative: “We report the main biological effects in relationship to autophagy regulation in response to different dietary polyphenols and its impact on metabolic and neurodegenerative diseases.” The authors should rewrite this sentence and add other 2 up to 3 sentences providing more information about the specific topics discussed in this work. The success of the review depends on the title and the abstract and a badly written summary will not attract the attention of potential readers.

- We have changed the abstract as suggested by the referee.

Lines 39 and 40. This sentence is still incorrect. If the authors do not want to mention other types of cell death, at least they should rephrase to “ The main response to tissue damage caused intense oxidative stress is cell death. The most studied type of cell death is apoptosis. It is runs through several steps, starting …”

- As suggested by the referee, we have modified the sentence accordingly (lines 46 and 47 of the fourth revised version of the manuscript)

The authors should shorten the description of apoptosis in the introduction. 6 lines is too much for a topic barely mentioned in the main text.

- We have diminished the description of apoptosis as recommended by the referee (lines 48-51 of the fourth revised version of the manuscript)

The section title: 4. Effect of polyphenols in cardiovascular complications” should be “… on cardiovascular diseases”, The term “complication” in medicine is restricted to “unwanted consequences of a primary disease”.

- The section title 4 have been changed as suggested by the referee

Line 54-55: The sentence “Furthermore, there are positive as well as negative regulatory loops in different proteins involved in either apoptosis or autophagy (16,17).” is unclear and out of context with respect to the previous sentence” It should be rephrased.

- We have rephrased the sentence, as suggested by the referee (lines 61-63 in the fourth version of the manuscript)

The manuscript is still rich in style and grammar errors. Just one example:

Line 60 and 221: “ .. increasing evidences demonstrate… ” should be “increasing evidence demonstrates”

- As suggested by the referee, we have corrected both sentences

The authors should carefully revise and eliminate all style and grammar errors, if they want to avoid further rounds of the revision.

- We have revised the whole manuscript in order to eliminate all the style and grammar errors